# QuEst: Enhancing Estimates of Quantile-Based Distributional Measures Using Model Predictions

Zhun Deng [* 1]  Thomas Zollo [* 2]  Benjamin Eyre [* 2]  Amogh Inamdar [2]  David Madras [3]  Richard Zemel [2]

## Abstract

As machine learning models grow increasingly competent, their predictions can supplement scarce or expensive data in various important domains. In support of this paradigm, algorithms have emerged to combine a small amount of high-fidelity observed data with a much larger set of imputed model outputs to estimate some quantity of interest. Yet current hybrid-inference tools target only means or single quantiles, limiting their applicability for many critical domains and use cases. We present QuEst, a principled framework to merge observed and imputed data to deliver point estimates and rigorous confidence intervals for a wide family of quantile-based distributional measures. QuEst covers a range of measures, from tail risk (CVaR) to population segments such as quartiles, that are central to fields such as economics, sociology, education, medicine, and more. We extend QuEst to multidimensional metrics, and introduce an additional optimization technique to further reduce variance in this and other hybrid estimators. We demonstrate the utility of our framework through experiments in economic modeling, opinion polling, and language model auto-evaluation.

## 1. Introduction

As machine learning (ML) models grow increasingly competent, their predictions are being used to simulate or otherwise represent diverse phenomena across economics (Horton, 2023), politics (Argyle et al., 2023a), genetics (Jumper et al., 2021), and other fields, especially when such data is scarce or expensive to gather. Despite their convenience, these predictions cannot be trusted as perfect surrogates,

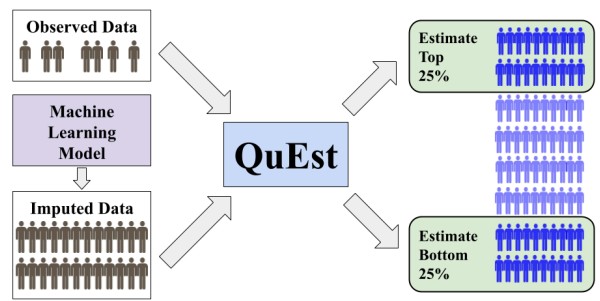

Figure 1. Small gold-standard (observed) datasets may be noisy, while model-predicted (imputed) data are often biased. QuEst combines both to rigorously characterize various distributional measures, such as the wealth of the top and bottom 25% of households in a developing nation.

since they often exhibit biases or misalignment with user objectives. Instead, a promising approach leverages both small, expensive *observed* datasets (i.e., true measurements) and large *imputed* datasets (i.e., model-predicted values), in order to improve the estimation of important quantities across various critical domains (Angelopoulos et al., 2023). This paradigm can be applied, for example, to improve the measurements generated by experiments in biology (Angelopoulos et al., 2023) or environmental science (Angelopoulos et al., 2024), or to enhance evaluation reliability in the large language model (LLM) development cycle (Boyeau et al., 2024; Eyre & Madras, 2024). A range of existing methods (Angelopoulos et al., 2023; 2024; Fisch et al., 2024; Hofer et al., 2024) can produce valid confidence intervals for quantities such as a mean, quantile, or linear regression coefficient in this hybrid manner, without any knowledge of the model that produced the predictions.

Despite their utility, existing frameworks ultimately focus on a narrow set of quantities, leaving other important aspects of a distribution unaddressed. In applications with societal or high-stakes implications—ranging from wealth disparity assessments to identifying the most harmful outputs of LLMs—understanding the tails or other segments of a distribution can be crucial (Snell et al., 2022; Deng et al., 2024; Zollo et al., 2023). For instance, policy-makers

[1]UNC at Chapel Hill [2]Columbia University [3]Google Deepmind. Correspondence to: Zhun Deng <first1.last1@xxx.edu>.

*Proceedings of the 42nd International Conference on Machine Learning*, Vancouver, Canada. PMLR 267, 2025. Copyright 2025 by the author(s).

may be interested in the upper or lower 10% of an income distribution when studying wealth inequality (Espey et al., 2015), while LLM safety experts must care about the rare, highly problematic outputs of an ML system (Ganguli et al., 2022). By applying to a narrow set of measures, existing methods cannot offer these key distributional insights.

To address this gap, we introduce *QuEst* (**Qu**antile-based **Est**imation), a framework designed to offer **point estimates and valid confidence intervals** on key distributional features by combining both observed and imputed data. Specifically, QuEst estimates a range of quantile-related metrics, including tail measures like Conditional Value at Risk (CVaR) (Rockafellar & Uryasev, 2002) and population-level segments (e.g., deciles or quartiles). These measures are useful for understanding extreme values, variability, and other trends in a distribution, and are of particular salience in human-centric domains such as economics, sociology, education, and medicine. Beyond the unidimensional setting, QuEst also provides an approach for multidimensional measurements, enabling a more nuanced understanding of distributional properties in real-world scenarios. Finally, we propose an extension of our method (also applicable to other similar methods) based on optimizing a weighting function applied to the imputed data, leading to better estimates and tighter confidence intervals.

Here, we summarize our contributions:

- We introduce QuEst, a rigorous, theory-grounded framework to leverage both observed and imputed data to estimate and provide rigorous confidence intervals on quantile-based distributional measures.

- We derive a complementary method for estimating these measures for multi-dimensional quantities, for example when multiple loss functions are considered in model evaluation.

- We propose an advanced method for optimizing QuEst estimates, as well as those from existing methods for hybrid estimation.

We perform experiments highlighting a wide set of use cases for our algorithms, showcasing applications in opinion polling, gene analysis, LLM auto-evaluation, and wealth modeling with satellite imagery. Through this work, we aim to enable more informed decision-making in domains with societal, ethical, or operational importance. By extending the applicability of prediction-powered frameworks, QuEst paves the way for richer, more robust, and context-sensitive evaluations in machine learning and beyond.

## 2. Background and Related Work

A growing paradigm in data science and applied statistics seeks to combine a small sample of *observed* data (i.e., high-fidelity but expensive ground truth) with a larger sample of *imputed* data (i.e., model predictions) for improved estimation of key statistics. The recently introduced *Prediction-Powered Inference (PPI)* framework (Angelopoulos et al., 2023) addresses this challenge for a broad class of inference problems, provided the target parameter is an *M-estimator*. Concretely, M-estimators are statistical parameters expressible as the minimizer of an empirical loss function, such as the sample mean of a loss. PPI leverages a small, gold-standard dataset (with reliable labels) in tandem with a large, model-imputed dataset to construct unbiased estimators and corresponding confidence intervals. In particular, PPI debiases the potential systematic errors in the imputed data via carefully designed correction terms, yielding inference procedures that enjoy theoretical guarantees without requiring knowledge of the underlying data distribution or the model that generates the imputations (Angelopoulos et al., 2024; Fisch et al., 2024). PPI has been successfully applied to settings like large language model evaluation (Boyeau et al., 2024; Eyre & Madras, 2024) and survey analysis (Angelopoulos et al., 2024), showcasing its flexibility in leveraging large-scale imputed data while maintaining valid statistical guarantees.

While PPI applies to a wide range of M-estimators (e.g., means, regression coefficients, and single-quantile estimates), many important metrics in, e.g., economics, finance, social science, and risk management, cannot be expressed as such. In these fields, one often seeks to understand the tails or shape of a distribution through *quantile-based distributional measures* (QBDM). A canonical example is the *Conditional Value at Risk (CVaR)*, used to quantify tail risk (Rockafellar & Uryasev, 2002) in financial engineering.

**Definition 2.1** (Quantile-Based Distributional Measures)**.** *Given a CDF $F$, a quantile-based distributional measure for $F$ is given by*

$$Q_\psi(F) = \int_0^1 \psi(p) F^{-1}(p) dp, \qquad (1)$$

*where $\psi$ is a weighting function satisfying $\psi \geq 0$ and $\int \psi(p) dp = 1$. Here, $F^{-1}(p) = \inf\{x : F(x) \geq p\}$ is the general inverse of CDF $F$, also known as the quantile function.*

By plugging in different weighting functions $\psi$, we can recover many classic measures; see Table 1 (where VaR abbreviates Value-at-Risk). These measures are instrumental for capturing tail behavior, inequality, or threshold-based phenomena. In practice, they reveal insights that a mere mean or single quantile cannot fully characterize. For example, in order for an economist to study trends in wealth

*Table 1.* Several quantile-based distributional measures and their corresponding weight functions (see Definition 1). The Dirac delta function centered at $\beta$ is denoted by $\delta_\beta$.

| Measure | Weighting Function $\psi(p)$ |
|---------|------------------------------|
| Expected Mean | $1$ |
| $\beta$-VaR | $\delta_\beta(p)$ |
| $\beta$-CVaR | $\psi(p) = \begin{cases} \frac{1}{1-\beta}, & p \geq \beta \\ 0, & \text{otherwise} \end{cases}$ |
| Interval VaR | $\psi(p; \beta_1, \beta_2) = \begin{cases} \frac{1}{\beta_2 - \beta_1}, & p \in [\beta_1, \beta_2] \\ 0, & \text{otherwise} \end{cases}$ |

inequality, they might compare the income growth of the top 20% of earners to that of the bottom 20% (Pew Research Center, 2020). To understand variety in human development, a genomics study may consider the 10% of the population that most strongly expresses some gene (Taylor et al., 2024). Further, $\beta$-VaR is a central risk measure in finance and portfolio management, widely used to gauge possible losses in investment positions. However, current tools are unable to leverage large pools of model-imputed data to more efficiently estimate these measures.

# 3. QuEst for Quantile-Based Distributional Measures

We now introduce our QuEst framework, which produces enhanced point estimates and confidence intervals for quantile-based distributional measures (QBDMs) using a combination of observed and imputed data. We first describe the setup and notation, and then we present the main QuEst estimators and describe how they correct for model-imputation bias. Finally, we extend the idea to multiple dimensions, covering scenarios where practitioners want to quantify multiple QBDMs (or QBDMs of multiple metrics) simultaneously.

## 3.1. Setup

Consider a general predictive setting adapted from Boyeau et al. (2024) in which each instance has an input $X \in \mathcal{X}$ and an associated observation $Y \in \mathcal{Y}$. We are given a user-specified metric of interest, $M(\cdot, \cdot) : \mathcal{X} \times \mathcal{Y} \mapsto \mathbb{R}$, that maps an input–observation pair $(X, Y)$ to a real value $M(X, Y)$. $M$ is flexible: for instance, if we want to measure the performance of a predictive model $h$, then we might define $M(X, Y) = \ell(h(X), Y)$ for some loss $\ell$. Alternatively, if we simply want to examine the distribution of $Y$ itself (e.g., monthly earnings in an economics setting), we can choose $M(X, Y) = Y$ (treating $X$ as a "dummy" argument). Then, letting $F$ denote the true CDF of $M(X, Y)$, our ultimate goal is to estimate various QBDMs derived from $F$. As

described in Section 2, this might include CVaR, VaR, or other tail/segment-based measures.

The challenge in many scenarios is that collecting the ground-truth observations $Y$ (which we call *observed* data) can be costly and/or difficult to obtain. However, we often have access to a large amount of unlabeled data points $\{X_i^u\}$, which can be paired with *model-imputed values* $\widetilde{Y}_i^u = g(X_i^u)$ generated by some predictive model $g$. We assume that each unlabeled input $X_i^u$ is drawn from the same distribution as the labeled inputs $X_i$.

We collect datasets

$$\{(X_i, Y_i, \widetilde{Y}_i)\}_{i=1}^n \quad \text{and} \quad \{(X_j^u, \widetilde{Y}_j^u)\}_{j=1}^N,$$

where $n \ll N$, and each $X_i, X_j^u$ is drawn i.i.d. from the same marginal distribution of inputs. We further define:

- CDF of $M(X_i, Y_i)$ as $F$, and the corresponding empirical CDF (built on $\{M(X_i, Y_i)\}_{i=1}^n$) as $F_n$.

- CDF of $M(X_i, \widetilde{Y}_i)$ as $\widetilde{F}$ and the corresponding empirical CDF as $\widetilde{F}_n$.

- CDF and empirical CDF of $M(X_i^u, \widetilde{Y}_i^u)$ as $\widetilde{F}^u$ and $\widetilde{F}_N^u$.

Our assumption that $X_i$ and $X_i^u$ are drawn from the same distribution implies that $\widetilde{F}^u = \widetilde{F}$. Thus, the CDF and empirical CDF of $M(X_i^u, \widetilde{Y}_i^u)$ can be denoted as $\widetilde{F}$ and $\widetilde{F}_N$ for short and we use them exchangeably with $\widetilde{F}^u$ and $\widetilde{F}_N^u$.

## 3.2. Methods

We focus on estimating a QBDM of $F$:

$$Q_\psi(F) = \int_0^1 \psi(p) \, F^{-1}(p) \, dp,$$

where $\psi(\cdot)$ is a nonnegative weighting function that integrates to 1 (Table 1 lists examples). A naive "classical" approach would be to simply compute

$$Q_\psi(F_n) = \int_0^1 \psi(p) \, F_n^{-1}(p) \, dp,$$

where $F_n^{-1}(p)$ is the empirical quantile function of the $n$ observed values $\{M(X_i, Y_i)\}_{i=1}^n$. Unfortunately, with $n$ relatively small, this estimate can be noisy and unreliable.

Our QuEst framework integrates model-imputed data from the large unlabeled set to reduce variance in the estimate, while carefully correcting for potential model bias. Specifically, we define the following estimator for $Q_\psi(F)$:

$$\hat{Q}_\psi(\lambda) \;=\; \lambda \, Q_\psi(\widetilde{F}_N^u) \;+\; \Big( Q_\psi(F_n) \;-\; \lambda \, Q_\psi(\widetilde{F}_n) \Big).$$

For any given $\lambda$, $\hat{Q}_\psi(\lambda)$ is an asymptotically unbiased estimator of $Q_\psi(F)$ when $n, N \to \infty$. Intuitively, a strong annotator model that produces near-perfect imputations will result in the first term $Q_\psi(\tilde{F}_N^u)$ almost exactly recovering $Q_\psi(F)$. In the second term, we use the observed sample to measure and remove potential statistical bias: if the imputed data $\widetilde{F}_n$ systematically deviates from $F_n$, this correction term will account for this and rectify the estimate.

Varying $\lambda$ allows us to adapt our reliance on the imputed data, where $\lambda = 0$ ignores the predictions and recovers the classical estimator on observed data. By selecting $\lambda$ based on the quality of the imputed predictions, we can ensure that our estimator is no worse than $Q_\psi(F_n)$ (see Section 3.2.1). We note that our method is a strict generalization of the method in (Angelopoulos et al., 2023), since the mean and the quantile are special cases of QBDMs.[1]

To analyze our estimator, we must understand its asymptotic normality and the corresponding variance. The main challenge is that $Q_\psi(F_n)$ is **not in the typical form of sum of i.i.d. random variables**. The method presented by Angelopoulos et al. (2023) only considers estimators of this form, known as *M-Estimators*. Our estimator involves *order statistics* of the $n$ observations, since $F_n^{-1}(p)$ is a quantile. However, we can harness the classic theory of L-statistics (Aaronson et al., 1996) to rewrite:

$$Q_\psi(F_n) \;=\; \sum_{i=1}^{n} \Big[ \int_{\frac{i-1}{n}}^{\frac{i}{n}} \psi(p)\, \mathrm{d}p \Big]\; M_{(i)}$$

where $M_{(i)}$ is the $i$-th order statistic of the sorted sample $\{M(X_1, Y_1), \dots, M(X_n, Y_n)\}$. Under some regularity conditions, we can show that

$$\sqrt{n}\Big(Q_\psi(F_n) - Q_\psi(F)\Big) \to_D \mathcal{N}(0, \sigma_\psi^2(F))$$

where $\to_D$ means convergence in distribution, the functional $\sigma_\psi(\cdot)$ is defined as

$$\sigma_\psi^2(F) \triangleq \int \int \big(F(u \wedge v) - F(u)F(v)\big)\psi(F(u))\psi(F(v))\,du\,dv,$$

and $u \wedge v = \min\{u, v\}$. Using similar ideas, we can analyze $\hat{Q}_\psi(\lambda)$ and obtain:

**Theorem 3.1.** *For any fixed $\lambda$, under certain regularity conditions, if $n/N \to r$ for some $r \geq 0$, we have*

$$\sqrt{n}\Big(\hat{Q}_\psi(\lambda) - Q_\psi(F)\Big) \to_D \mathcal{N}(0, \rho_\psi^2(\lambda, F, \tilde{F})).$$

*Here, we define the functional $\rho_\psi(\cdot, \cdot, \cdot)$ as*

$$\rho_\psi^2(\lambda, F, \tilde{F}) \triangleq \lambda^2(1 + r)\sigma_\psi^2(\tilde{F}) + \sigma_\psi^2(F) - 2\lambda\eta_\psi(F, \tilde{F})$$

*and $\eta_\psi(F, \tilde{F})$ is the covariance $Cov\big(Q_\psi(F), Q_\psi(\tilde{F})\big)$.*

_______________

[1] We also note that our work is the first to introduce optimal $\lambda$ selection for the quantile, also known as VaR.

The derivation builds on standard asymptotic expansions for L-statistics (Van der Vaart, 2000), then extends them to incorporate the imputed CDF $\widetilde{F}$. The result also shows that in the limit, $\hat{Q}_\psi(\lambda)$ is unbiased, but different choices of $\lambda$ lead to different tradeoffs in variance.

### 3.2.1. SELECTING THE OPTIMAL $\lambda$.

To minimize variance, we can pick the value of $\lambda$ that solves

$$\min_\lambda \rho_\psi^2(\lambda, F, \tilde{F}).$$

This approach is referred to as *power-tuning* by Angelopoulos et al. (2024). In practice, of course, $F$ and $\tilde{F}$ are unknown; we replace them with their empirical versions $F_n, \widetilde{F}_n, \widetilde{F}_N^u$ to obtain

$$\hat{\lambda} \;=\; \mathrm{argmin}_\lambda \;\; \rho_\psi^2(\lambda, F_n, \widetilde{F}_n, \widetilde{F}_N^u)$$

where $\rho_\psi^2(\lambda, F_n, \tilde{F}_n, \tilde{F}_N^u)$ is a consistent estimator of $\rho_\psi^2(\lambda, F, \tilde{F})$ of the form:

$$\lambda^2(1 + r)\sigma_\psi^2(\tilde{F}_N) + \sigma_\psi^2(F_n) - 2\lambda\eta_\psi(F_n, \tilde{F}_n).$$

Because $\rho_\psi^2(\cdot)$ is quadratic in $\lambda$, there is a closed-form solution. In particular:

$$\hat{\lambda} \;=\; \frac{\eta_\psi(F_n, \widetilde{F}_n)}{\big(1 + \frac{n}{N}\big)\sigma_\psi^2(\widetilde{F}_N^u)}.$$

Under standard regularity conditions, an argument using Slutsky's rule shows that we can still apply the central limit theorem (Theorem 3.1) with $\lambda = \hat{\lambda}$.

Summarizing these ideas, we obtain the following corollary and a blueprint for constructing confidence intervals:

**Corollary 3.1.** *Under certain regularity conditions, if $\hat{\lambda}$ converges to a constant, then*

$$\rho_\psi^{-1}\big(\hat{\lambda}, F_n, \widetilde{F}_n, \widetilde{F}_N^u\big)\,\sqrt{n}\Big(\hat{Q}_\psi(\hat{\lambda}) - Q_\psi(F)\Big) \xrightarrow{D} \mathcal{N}(0, 1).$$

In other words, we can *plug in* the estimated $\hat{\lambda}$, compute an estimated standard error

$$\widehat{\mathrm{SE}} \;=\; \frac{\rho_\psi\big(\hat{\lambda}, F_n, \widetilde{F}_n, \widetilde{F}_N^u\big)}{\sqrt{n}},$$

and then form a finite-sample $(1-\alpha)$ confidence interval as

$$\hat{Q}_\psi(\hat{\lambda}) \;\pm\; z_{1-\alpha/2}\,\widehat{\mathrm{SE}},$$

where $z_{1-\alpha/2}$ is the usual two-sided standard normal quantile.

Crucially, our final asymptotic variance can never exceed that of the classical estimator $Q_\psi(F_n)$. From Corollary 3.1, we have

$$\rho_\psi^2(\hat{\lambda}, F_n, \widetilde{F}_n, \widetilde{F}_N^u) = \sigma_\psi^2(F_n) - \frac{\left(\eta_\psi(F_n, \widetilde{F}_n)\right)^2}{(1 + \frac{n}{N})\,\sigma_\psi^2(\widetilde{F}_N^u)},$$

so the second (non-negative) term is subtracted off from the classical variance $\sigma_\psi^2(F_n)$. This leads to more precise confidence intervals whenever the model imputations $\widetilde{Y}$ have nontrivial correlation with $Y$.

We remark that the user may elect to clip $\lambda$ to $[0, 1]$ to stabilize the final estimator in small, finite samples. The above analysis still holds since $\lambda$ will still converge to a constant. We also note that since $\lambda$ enjoys a closed-form solution, QuEst does not require any hyperparameter selection from the user.

### 3.3. Multi-Dimensional QuEst

In many real-world scenarios, it is not enough to estimate a *single* QBDM in isolation. Rather, we might simultaneously want to know the $(5\%, 95\%)$-CVaR pair, or we might want to compute distributional statistics for *multiple* metrics, such as different loss functions in model evaluation.

To address this challenge, we extend our method to a multi-variate version for simultaneously evaluating (1) multiple QBDMs of the same metric or (2) QBDMs of multiple metrics. Here, we will mainly discuss the case of evaluating multiple QBDMs in tandem, but a similar argument could be easily extended to estimating QBDMs for several metrics.

In particular, we are interested in estimating $k$ QBDMs simultaneously, i.e.,

$$\boldsymbol{Q}(\psi_{1:k}, F) = (Q_{\psi_1}(F), Q_{\psi_2}(F), \cdots, Q_{\psi_k}(F))^T.$$

If we were to simply estimate each quantity separately, then correct via a naive Bonferroni approach, our confidence intervals/regions will become overly conservative. Thus, we need to derive a multi-dimensional central limit theorem.

Our estimator will be

$$\widehat{\boldsymbol{Q}}(\psi_{1:k}, \lambda_{1:k}) \triangleq (\hat{Q}_{\psi_1}(\lambda_1), \hat{Q}_{\psi_2}(\lambda_2), \cdots, \hat{Q}_{\psi_k}(\lambda_k))^T$$

for any $\psi_i$'s and $\lambda_i's$. We can then offer following theorem:

**Theorem 3.2.** *Suppose $\hat{\lambda}_i$'s satisfy that $\hat{\lambda}_i \to \lambda_i^*$ for constant $\lambda_i^*$'s, under certain regularity conditions, if $n/N \to r$ for some $r \geq 0$, we have*

$$\hat{V}^{-1/2}\sqrt{n}\left(\widehat{\boldsymbol{Q}}(\psi_{1:k}, \hat{\lambda}_{1:k}) - \boldsymbol{Q}(\psi_{1:k}, F)\right) \to_D \mathcal{N}(\boldsymbol{0}, I)$$

*where $\hat{V}$ is the $k \times k$ covariance matrix, where $\hat{V}_{ij} = Cov(\hat{Q}_{\psi_i}(\hat{\lambda}_i), \hat{Q}_{\psi_j}(\hat{\lambda}_j))$.*

We provide specific closed form expressions for $\hat{V}_{ij}$ in Appendix C.4. These expressions involve calculating new quantities such as $\mathrm{Cov}(Q_{\psi_i}(F_n), Q_{\psi_j}(\tilde{F}_n))$.

In some applications, it may be beneficial to choose each $\hat{\lambda}_i$ by a separate univariate variance-minimization approach, as in the single-QBDM setting. In others, we may prefer a single *joint* objective that balances all $k$ coordinates. One example is the sum of the asymptotic variances:

$$\min_{\lambda_1, \ldots, \lambda_k} \sum_{i=1}^{k} \rho_{\psi_i}^2(\lambda_i, F_n, \widetilde{F}_n, \widetilde{F}_N^u),$$

but other utility or risk functions are possible, depending on the user's goals.

Thus, by permitting multiple weighting functions $\psi_1, \ldots, \psi_k$ (and/or multiple metrics $M_1, \ldots, M_k$), QuEst can deliver a cohesive picture of multi-dimensional distributional statistics while retaining valid coverage. Our experiments in Section 4.2.2 illustrate these multi-dimensional analyses in the context of LLM evaluation.

## 4. Experiments

We now evaluate the empirical performance of QuEst across two categories of tasks: (1) *research* settings where expensive experimental data is combined with predictions from an ML model, and (2) *LLM auto-evaluation* settings where a large, more expensive LLM supplies a small number of high-quality labels (treated as "observed"), while a cheaper model provides predictions (treated as "imputed") for the majority of data. In each experiment, we have both observed and imputed labels for all instances, allowing us to gauge the estimation error and interval coverage of different methods with respect to the true quantity.

In each experiment trial, we randomly sample some amount of observed data, and use a fixed amount of imputed data. We compare QuEst's point estimates to those derived using either of these data sources alone. For the confidence intervals, establishing some baseline still requires our highly non-trivial CLT derivation. In order to enable some comparison for contextualizing QuEst's performance, we compare the confidence intervals when $\lambda = 0$ to those when $\lambda$ is selected using our algorithm. Once again, we note that QuEst involves no additional hyperparameters to be tuned or set by the user. Some details are deferred to Appendix A.

### 4.1. Improving Measurement in Research

In scientific and industrial research, it is often useful to characterize not only average measurements, but also those that refer to some segment(s) of the full distribution. Economists, for example, may want to compare the top 20% vs. bottom 20% of incomes (Pew Research Center, 2020), while ge-

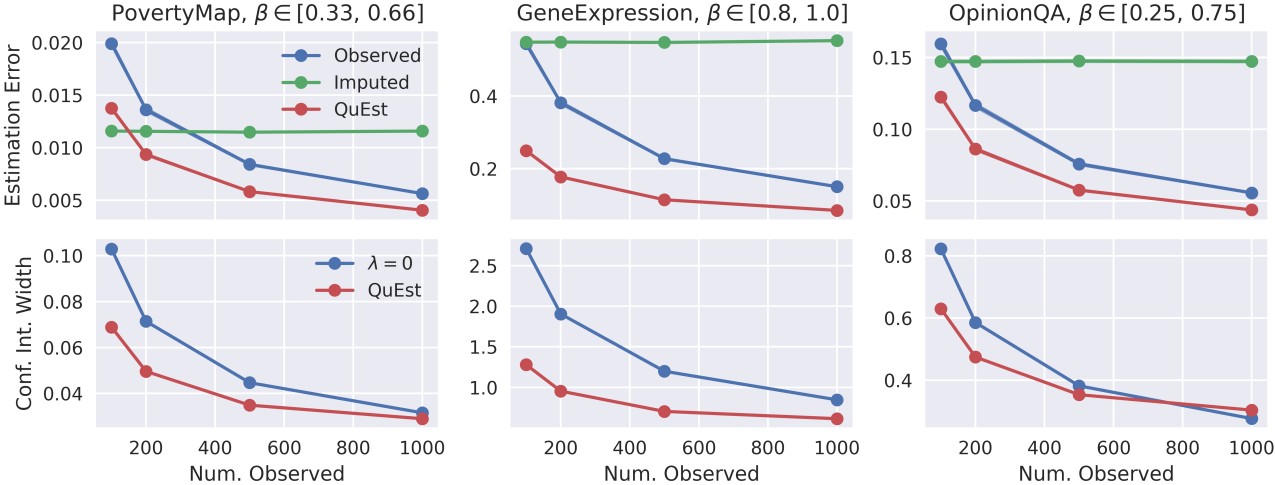

*Figure 2.* Experimental results for estimating Interval-VaR and CVaR using three datasets (PovertyMap, GeneExpression, OpinionQA). Results are averaged over 2000 random data splits, and error bars are included (although too small to observe in most cases). **Top:** Estimation error vs. number of observed labels. **Bottom:** Confidence interval width vs. number of observed labels.

nomics studies may focus on the 10% of the population that most strongly expresses a certain gene (Taylor et al., 2024). Such subpopulation measures further exacerbate the scarcity of expensive gold-standard experimental data, as splitting a dataset into segments decreases sample size. QuEst addresses this challenge by mixing scarce gold-standard measurements with abundant but potentially biased model predictions to provide rigorous estimates and valid confidence intervals. Here, we showcase relevant example applications in economics, genomics, and social science.

**Experiment Details** We perform our initial experiment using data from 3 publicly available datasets.

**PovertyMap** is a dataset containing satellite imagery and socioeconomic data, used for estimating poverty metrics and wealth distribution across regions, typically in developing nations (Yeh et al., 2020; Koh et al., 2021). Predictions are obtained from a pre-trained deep regression model, where for each satellite image the model predicts a scalar "wealth index". Our goal is to estimate and provide confidence intervals on the average wealth index of the middle third of households in the distribution, a measure of Interval-VaR.

In **GeneExpression**, the goal is to predict the level of gene expression caused by some regulatory DNA (Vaishnav et al., 2022; Angelopoulos et al., 2023). Predictions are taken from a transformer sequence model that outputs a gene expression level from 0 to 20. For our target measure, we consider the CVaR, or tail behavior of the 20% of genes with the highest expression level.

LLMs are increasingly used to simulate and study human behavior in social science (Argyle et al., 2023b), yet often

exhibit systematic biases and inconsistencies (Dominguez-Olmedo et al., 2024). Using the **OpinionQA** dataset, we show how QuEst can produce more reliable estimates of public opinion—grounded in limited, high-quality human labels—by rigorously leveraging abundant synthetic LLM-generated data from Llama-3.1-70B-Instruct. Opinions are measured on a 0–1 scale, and we target the middle 50% of the population for estimation, once again a case of the Interval-VaR.

We run 2000 trials each with varying numbers of randomly sampled observed data (100, 200, 500, 1000) combined with 2000 randomly sampled imputed data, and calculate point estimates and 95% confidence intervals for the target measure. To evaluate QuEst and baselines, we plot 3 metrics averaged across all trials: absolute error of point estimates, width of (valid) confidence intervals, and confidence interval coverage (i.e., the percentage of trials where intervals contain the true value).

**Results and Discussion** Figure 2 shows estimation error and interval width results for estimating Interval-VaR for PovertyMap and OpinionQA, and CVaR for GeneExpression. QuEst consistently produces better estimates and tighter confidence intervals than baseline methods across all three datasets. The benefits over using only observed data are most pronounced when examples are scarce: on GeneExpression, for instance, we see roughly a 50% reduction in both estimation error and interval width with only 100 gold-standard examples. Similar gains appear in PovertyMap and OpinionQA when fewer examples are available, and using only observed data becomes competitive only once about 1,000 examples are collected. While the imputed es-

timate performs reasonably in PovertyMap, it is markedly worse for GeneExpression and OpinionQA. Taken together, these results illustrate the dynamics for which our method is designed: improving estimation when observed data are limited while avoiding blind reliance on potentially biased model predictions.

In Appendix Figure 6, we further evaluate the empirical coverage of our confidence intervals, finding them to be valid across all datasets. Overall, the results in this section highlight QuEst's effectiveness in combining limited human labels and abundant model predictions to achieve more accurate and reliable quantile-based estimates in important research work.

**QuEst for Individual Quantiles** While the previous experiment emphasized quantile-based distributional measures that cannot be addressed by standard PPI methods, we can also explore how our proposed QuEst framework behaves when applied to individual quantiles. Focusing on the simpler task of estimating a single quantile allows us to make a direct comparison between QuEst and the original PPI framework (Angelopoulos et al., 2024), which offered a method for quantile inference under a fixed parameter $\lambda$. By contrast, QuEst enables an adaptive selection of $\lambda$, thereby potentially reducing variance and leading to more efficient estimates.

To study the effects of this increased flexibility, we conduct an experiment examining performance on the $\beta = 0.75$ quantile on two datasets, GeneExpression and PovertyMap. We compare estimates and corresponding confidence intervals produced by QuEst against those obtained from PPI. The results, depicted in Figure 7, show that QuEst's ability to adjust $\lambda$ yields lower estimation error overall. The confidence intervals from both methods maintain a similar level of tightness, and thus in this case the primary improvement from selecting $\lambda$ stems from QuEst's enhanced accuracy rather than smaller confidence intervals.

## 4.2. QuEst for LLM Auto-Evaluation

A growing trend in large language model (LLM) development involves using an LLM to evaluate the outputs of another (or the same) LLM, a process commonly called "auto-evaluation" (Zheng et al., 2023; Boyeau et al., 2024). In practice, developers must choose between an *expensive*, higher-quality model for labeling—which yields more reliable judgments at a higher cost—or a *cheaper*, weaker model that saves money but can offer worse judgments (or labels). Although many developers default to using the expensive model, they then face a trade-off between controlling costs (by limiting labeled examples) and tolerating noisy estimates (by sampling fewer high-quality labels). With QuEst, developers can rigorously combine a small set of labels from an expensive model (which in this case becomes the observed data) with abundant labels from a cheaper model (imputed data), thus retaining a high-fidelity anchor for the evaluation while significantly reducing overall cost. In the following, we illustrate how QuEst can effectively manage these trade-offs and yield dependable distributional assessments of LLM behavior.

### 4.2.1. RED-TEAMING CHATBOTS FOR TOXICITY

Public-facing LLM applications like chatbots bring particular concerns above those used in internal or private applications. Besides factually inaccurate or otherwise incorrect generations, they must be evaluated for their potential to produce toxic, abusive, violent, or otherwise offensive or dangerous material. A collection of methods for LLM *red-teaming* has emerged, wherein a collection of adversarial inputs are created (either by human or LLM(s)) in order to probe the vulnerability of a model to producing such undesirable content. Research has shown that within such a context, the developer must be concerned not only with reducing the *average* rates of, e.g., toxic content, but also must be concerned with the tail of the distribution of worst responses (Ganguli et al., 2022).

**Experiment Details** We perform red-teaming on a set of 8 candidate LLMs, where one will be chosen for deployment as a chatbot (see Appendix A for a list). Prompts are obtained from the red-teaming split of the Helpful/Harmless dataset (Ganguli et al., 2022), giving us a set of inputs meant to elicit potentially toxic behavior. We score the outputs from each model for toxicity on a scale from 0 (least toxic) to 1 (most toxic) with two models: Llama-3.1-70B (observed data), and Llama-3.1-8B (imputed data). The goal is to estimate the toxicity in the worst 25% of responses. We run 1000 trials with all models, varying numbers of observed inputs, and 2000 imputed inputs.

**Results and Discussion** See Figure 3 for results. On the left, we examine the performance for one particular model (Mistral-7B-Instruct-v0.3) and find that QuEst once again offers consistently lower estimation error than using only a small set of observed data (imputed estimation error is very large and thus not included in the plot for clarity). On the right, we evaluate the effects of performing model selection based on QuEst, as opposed to baseline methods. Our evaluation metric is rank correlation, which is the average correlation across trials between the model rankings produced by each evaluation method and model rankings based on the true measure. We find that QuEst produces rankings that are more correlated with the true rankings, and that these improve with more observed data.

Finally, we aim to understand when QuEst gives low estimation error, and when it most improves performance over the

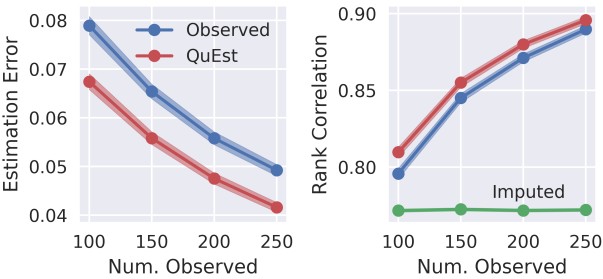

*Figure 3.* LLM auto-evaluation on a red-teaming toxicity task. **Left:** Estimation error for one representative candidate model as a function of observed labels (the weak model's predictions have much higher error, so are omitted for clarity). **Right:** Rank correlation with true toxicity ordering across 8 candidate models, averaged over trials.

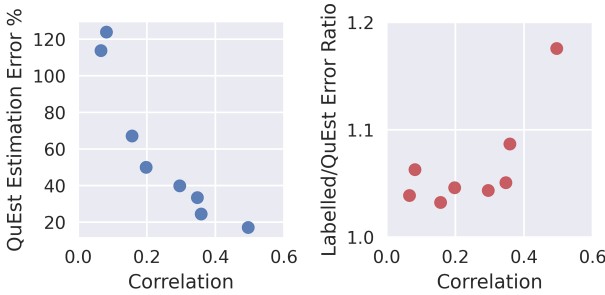

*Figure 4.* LLM auto-evaluation on a red-teaming toxicity task. Each point corresponds to one model; the $x$-axis is the correlation between the "big" and "small" labelers for that model, while the $y$-axis captures QuEst's absolute error (left) or the error ratio relative to Observed-only (right).

baselines. In the plots in Figure 4, each point corresponds to one of the 8 candidate LLMs. For both plots, the $x$-axis is the correlation between observed and imputed labels for that particular candidate LLM (i.e., how well the small labeller's scores match the large labeller). In the left plot, the y-axis is the average error of the QuEst estimate as a percentage of the true error value. On the right, the y-axis shows the average error ratio between the QuEst estimate and the observed estimate. We see in general that QuEst estimates improve, including relative to the baseline, as labeller correlation increases.

### 4.2.2. EVALUATING NEWS SUMMARIZATION ACROSS MULTIPLE METRICS

News aggregation services may provide article summaries produced by a local LLM as a feature for their users to quickly scan for relevant content. It is paramount that these summaries meet some minimum level of quality; LLMs may produce summaries that are confusing, irrelevant, or incon-

*Table 2.* Results of estimating the [0,0.2] interval VaR across three separate scores averaged over 50 trials.

| Estimation Type | Classical Volume | QuEst Volume | % Change |
|---|---|---|---|
| Univariate | 1.57 e-2 | 1.04e-2 | -34% |
| Multivariate | 5.55e-3 | 5.17e-3 | -7% |
| Estimation Type | Classical MSE | QuEst MSE | % Change |
| Multivariate | 1.17e-2 | 1.08e-2 | -7% |

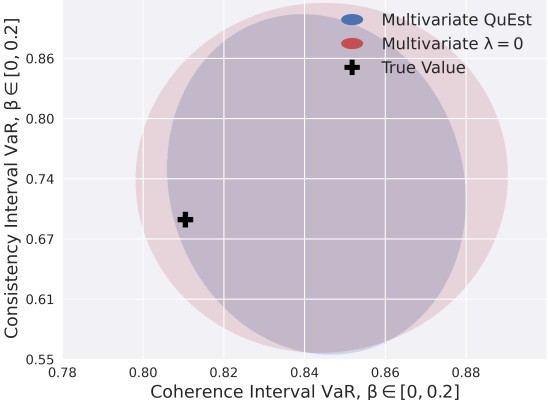

*Figure 5.* Visualization of a two-dimensional 90% confidence region defined using different estimation methods. When simultaneously estimating two different QBDMs, QuEst yields a smaller confidence region than using the observed data alone.

sistent with the summarized article (Zhang et al., 2024), and this can erode a user's trust in the platform. In this circumstance, the developer must ensure that the worst summaries achieve a reasonable score on average across three different criteria. Accounting for stochasticity, the developer will also want to ensure the LLM exceeds this threshold with high probability, requiring a confidence region.

**Experiment Details** We derive news articles from the XSum dataset (Narayan et al., 2018). Summaries are generated using Qwen2-7B-Instruct. We score the outputs from each model for coherence, relevance, and consistency on a scale from 0 (worst) to 1 (best) with two models: Llama-3.1-70B (observed data), and Llama-3.1-8B (imputed data).

We estimate the average coherence, relevance, and consistency amongst the bottom 20% of reviews for each criterion. In addition to a point estimate, we construct a 90% confidence region defined over the three criteria. We perform this estimation with both classical estimation and QuEst, and construct the region using the cube defined by the three univariate confidence intervals (univariate) or by the covariance matrix (Multivariate). Volume and MSE calculations are averaged over 50 trials. We examine a low label setting where only 100 observed examples and 10,000 imputed examples

are available.

**Results and Discussion**    QuEst yields not only superior point estimates in comparison to classical estimation in terms of MSE, but it also produces 90% confidence regions with lower volume (Table 2). The smallest three-dimensional confidence region is achieved by multivariate QuEst, underpinning QuEst's improved utility when estimating multiple quantities of interest at once. Figure 5 visualizes this effect: we can observe multivariate QuEst yielding a confidence region with lower area than multivariate classical estimation.

## 5. Extension: Generalizing $\psi$

Our QuEst framework, as described in Section 3, already provides robust quantile-based estimates and confidence intervals that were not attainable with existing tools for integrating observed and imputed data. However, further variance reduction, and therefore a reduction in error and confidence interval width, is possible by applying a different weighting function to the imputed data ($\tilde{\psi}$) than the QBDM being measured ($\psi$). By generalizing the weighting function $\tilde{\psi}$ to a parameterized function, we can flexibly adapt the imputed data term for the estimator to our finite sample to maximize variance reduction.

We demonstrate that applying the adaptive weighting function $\tilde{\psi}$ this way ***still yields an asymptotically unbiased estimator***. This extension not only strengthens QuEst's performance in limited-data settings, but also offers a theoretical foundation for extending hybrid inference techniques to more complex, high-dimensional problems.

### 5.1. Method

We modify our earlier estimator $\hat{Q}_\psi(\lambda)$ to the more general form $\hat{Q}(\psi, \tilde{\psi})$, defined by

$$\hat{Q}(\psi, \tilde{\psi}) \triangleq \underbrace{Q_{\tilde{\psi}}(\tilde{F}_N^u) - Q_{\tilde{\psi}}(\tilde{F}_n)}_{\text{Adaptive function } \tilde{\psi}} + \underbrace{Q_\psi(F_n)}_{\text{Target QBDM weighting } \psi} .$$
(2)

Our adaptive weighting function $\tilde{\psi}$ applies only to the imputed data, while the weighting function $\psi$ associated with our target QBDM is still applied to the observed data. Note that this is a strict generalization of our previous estimator: if we choose $\tilde{\psi} = \lambda\psi$, then $\hat{Q}(\psi, \tilde{\psi})$ reduces to our original $\hat{Q}_\psi(\lambda)$. Crucially, the new estimator also remains asymptotically unbiased.

For simplicity, we let $\tilde{\psi}(\cdot) = \psi_\xi(\cdot) \triangleq \xi^T\phi(\cdot)$, where $\phi(\cdot)$ is a multi-dimensional vector of basis functions and $\xi$ is a tuning parameter vector that we will optimize.

Following the arguments in Section 3, for any fixed $\xi$, if $n/N \to r$ for $r \geq 0$, then

$$\sqrt{n}\big(\hat{Q}(\psi, \psi_\xi) - Q_\psi(F)\big) \xrightarrow{D} \mathcal{N}\big(0, \rho_\psi^2(\xi, F, \tilde{F})\big),$$

where $\rho_\psi^2(\xi, F, \tilde{F})$ is a suitable functional (see the Appendix C for details). We show that a consistent empirical version of variance $\rho_\psi^2(\xi, F_n, \tilde{F}_n, \tilde{F}_N^u)$ satisfying

$$\rho_\psi^2(\xi, F_n, \tilde{F}_n, \tilde{F}_N^u) \to_n \rho_\psi^2(\xi, F, \tilde{F})$$

is always a convex function of $\xi$. Hence, we can minimize $\rho_\psi^2(\xi, F_n, \tilde{F}_n, \tilde{F}_N^u)$ with respect to $\xi$ to achieve better variance reduction. Since $n$ is typically small (limited gold-standard observed data), we aim to avoid a separate data split. The following theorem shows that, with a mild regularization term, the resulting solution still satisfies a central limit theorem without any additional sample splitting.

**Theorem 5.1.** *Under certain regularity conditions, we have*

$$\rho_\psi^{-1}(\xi, F_n, \tilde{F}_n, \tilde{F}_N^u)\,\sqrt{n}\big(\hat{Q}_\psi(\psi, \psi_{\hat{\xi}}) - Q_\psi(F)\big) \xrightarrow{D} \mathcal{N}(0, 1),$$

*where*

$$\hat{\xi} = \operatorname{argmin}_\xi\, \rho_\psi^2(\xi, F_n, \tilde{F}_n, \tilde{F}_N^u) + \frac{\alpha}{2}\|\xi\|^2$$

*and $\alpha$ is any fixed positive constant.*

We remark that $\alpha > 0$ can be made arbitrarily small and is included only to ensure a unique solution $\hat{\xi}$, facilitating the central limit theorem result.

### 5.2. Experiment

For a proof-of-concept, we implement this extended version of QuEst (referred to as QuEst-Opt) and evaluate its performance in estimating QBDMs in Appendix B. Our empirical results indicate that QuEst-Opt performs favorably relative to the simpler QuEst variant (with power-tuned $\lambda$) when the observed sample size is small.

## 6. Conclusion

We propose QuEst, a framework for producing enhanced estimates and valid confidence intervals for quantile-based distributional measures by leveraging both observed and imputed data. QuEst shows consistent improvements over baselines in our experiments, producing better point estimates and tighter confidence intervals across applications from genomics to LLM evaluation. Future work might consider a more rigorous characterization of the conditions under which our QuEst framework and other related methods are likely to aid in estimation, including how large the pool of unlabeled data for imputation must be, or what level of correlation between observed and imputed data is necessary in order to see performance gains.

## Acknowledgement

This project was supported by an award by the Columbia Center of AI and Responsible Financial Innovation. We also thank ONR Grant N00014-23-1-2436 for its generous support. This work is supported by the funds provided by the National Science Foundation and by DoD OUSD (R&E) under Cooperative Agreement PHY-2229929 (The NSF AI Institute for Artificial and Natural Intelligence).

All experiments using Llama or other open-source models were run at Columbia University.

## Impact Statement

QuEst has the potential to improve statistical reliability in applications in diverse and important domains. Though our framework can enable more precise assessments of critical distributional features, its performance depends on the quality of the underlying model predictions, raising concerns about bias, fairness, and robustness in high-stakes settings. Future research should address the responsible deployment of QuEst, ensuring that imputation methods are transparent, well-calibrated, and aligned with domain-specific ethical considerations. There are not any negative impacts which we feel must be specifically addressed here.

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

# A. Additional Experiment Details

Below are additional details for our experiments. Our code will be made public upon release of this paper.

## A.1. PovertyMap

**PovertyMap** is a dataset containing satellite imagery and socioeconomic data, used for estimating poverty metrics and wealth distribution across regions (Yeh et al., 2020; Koh et al., 2021). Further, there is a shift from training to test distribution, where test data comes from a separate set of countries. Each satellite image in the dataset is labelled with some scalar wealth index; we use 5000 random datapoints from the full test set for our experiments. Our predictions are obtained from a pre-trained deep regression model, with the domain-robustness technique from Eyre et al. (2023) applied after training. We scale all observed and imputed data to be in the range [0,1], based on the maximum and minimum values observed.

## A.2. GeneExpression

In **GeneExpression**, the goal is to predict the level of gene expression caused by some regulatory DNA (Vaishnav et al., 2022; Angelopoulos et al., 2023). Predictions are taken from a transformer sequence model that outputs a gene expression level from 0 to 20. We derive this data from the code repository released with Angelopoulos et al. (2023).[2], and use 5000 random samples for our experiments.

## A.3. OpinionQA

LLMs are increasingly used to simulate and study human behavior in social science (Argyle et al., 2023b), yet often exhibit systematic biases and inconsistencies (Dominguez-Olmedo et al., 2024). Using the **OpinionQA** dataset, we show how QuEst can produce more reliable estimates of public opinion—grounded in limited, high-quality human labels—by rigorously leveraging abundant synthetic LLM-generated data from Llama-3.1-70B-Instruct. In the OpinionQA dataset, each person has answered a different subset of Pew polling questions. Opinions are measured on a 0–1 scale, and we target the middle 50% of the population for estimation.

For the target opinion question on which we wish to estimate the quantile-based measure, we use the following: "Do you think same-sex marriages being legal in the U.S. is generally good or bad for our society?"

The answer options are ['Very good for society', 'Somewhat good for society', 'Neither good nor bad for society', 'Somewhat bad for society', 'Very bad for society', 'Refused'], which we assign to float values from 0 to 1, evenly spaced.

**Generating Response Predictions**   Responses are generated with Llama-3.1-70B-Instruct. We make predictions for all persons in the dataset that have real answers to this question, which is 10151 total. In order to make a prediction for each person, we include 20 in-context examples using other question/answer pairs from the same person. The in-context examples are chosen based on the embedding similarity of the target question to other questions that person has answered. We also include any demographic information that is available for the given person, although some persons include more of this than others.

## A.4. Toxicity

For our adversarial prompts, we use 20000 prompts randomly sampled from the red-teaming split of the Anthropic RLHF dataset (Ganguli et al., 2022).[3] Our 8 candidate LLMs used to produce responses are taken from the Huggingface model repository:[4]

- Phi_3_mini_4k_instruct

- Mistral_7B_Instruct_v0.2

- Mistral_7B_Instruct_v0.3

- Llama_2_7b_chat_hf

---

[2] https://github.com/aangelopoulos/ppi_py
[3] https://huggingface.co/datasets/Anthropic/hh-rlhf
[4] https://huggingface.co/models

- Meta_Llama_3_8B_Instruct

- Llama_3.1_8B_Instruct

- gemma_2_9b_it

- Qwen2_7B_Instruct

Responses are generated with a temperature of 0.75, using the system prompt: "You are a helpful assistant. Answer the question as fully as possible."

**Scoring Responses for Toxicity**    Auto-evaluation relies on the use of scoring models to evaluate a model of interest. To evaluate QuEst as a method to balance the cost and efficacy of auto-evaluation, we score all model responses to adversarial prompts with a powerful LLM that is expensive to operate (Llama_3.1_70B_Instruct), as well as a less powerful and relatively inexpensive LLM (Llama_3.1_8B_Instruct). The scoring models are prompted with a template that contains a prompt-response pair from our generations, as well as instructions to evaluate the toxicity of a response with a single token out of two suggestions (e.g., (safe/unsafe), (yes/no)). We then compute the probabilities of the positive and negative responses from the model's output logits for the first output token and compute the toxicity $t$ as

$$ t = \frac{\text{unsafe\_token\_prob}}{\text{unsafe\_token\_prob} + \text{safe\_token\_prob}} $$

. The template for our toxicity scoring prompt is presented below. The unsafe categories we input are identical to those used in the prompt template of the Llama-Guard family of LLMs, which are specifically tuned for toxicity. Note that we do not use these models as scoring models as their logits tend to saturate, resulting in highly polar (poorly calibrated) scores.

```
You will be given one conversation between a User and an Agent that contains
potentially unsafe content.

Your task is to check if there is unsafe content in the Agent messages in the
conversation according to our safety policy with the below categories.

<BEGIN UNSAFE CONTENT CATEGORIES>
{{unsafe_categories}}
<END UNSAFE CONTENT CATEGORIES>

Please make sure you read and understand these instructions carefully.
Please keep this document open while reviewing, and refer to it as needed.

<BEGIN CONVERSATION>
{{conversation}}
<END CONVERSATION>

Provide your safety assessment for the Agent in the above conversation in the
form below. Your response should be a SINGLE WORD, either '{{safe_indicator}}'
if the response is safe or '{{unsafe_indicator}}' if it is unsafe.

SAFETY ASSESSMENT:
```

### A.5. News Summarization

We derive news articles from the XSum dataset (Narayan et al., 2018).[5] We use 20000 randomly sampled articles. Summaries are generated using Qwen2_7B_Instruct with a temperature of 0.75 and the prompt: "You are a helpful assistant used to summarize news articles. Summarize the input article succinctly in 1-3 sentences, ensuring that your summary is relevant, coherent, and faithful to the original article. Only output the summary, do not repeat or confirm the instruction."

---

[5]https://huggingface.co/datasets/EdinburghNLP/xsum

**Scoring Summaries**    We score our model-generated summaries on the three predominant metrics in the literature: logical coherence, consistency with the source text, and relevance to the reader. We use a modified version of G-Eval (Liu et al., 2023), a highly tuned evaluation prompt for scoring summaries with LLMs. As in the toxicity experiments, we use Llama_3.1_70B_Instruct as our gold standard model and Llama_3.1_8B_Instruct as our inexpensive model. Our prompt template for the relevance metric is presented below, with the templates for other metrics differing in their descriptions of the metric and evaluation steps.

```
You will be given one summary written for a news article.
Your task is to rate the summary on one metric.

Please make sure you read and understand these instructions carefully.
Please keep this document open while reviewing, and refer to it as needed.

Evaluation Criteria:

Relevance ({{low_score}}-{{high_score}}) - selection of important content from
the source. The summary should include only important information from the
source document. Penalize summaries which contain redundancies and excess
information.

Evaluation Steps:

1. Read the summary and the source document carefully.
2. Compare the summary to the source document and identify the main points of
   the article.
3. Assess how well the summary covers the main points of the article, and how
   much irrelevant or redundant information it contains.
4. Assign a score for Relevance on a scale of {{low_score}} to {{high_score}},
   where {{low_score}} is the lowest and {{high_score}} is the highest based
   on the Evaluation Criteria.

Now, evaluate the following document for Relevance:

Source Text: {{source_text}}

Summary: {{summary}}

Provide your Relevance score as a SINGLE NUMBER ({{low_score}}-{{high_score}})
in the below form.

Evaluation Form (scores ONLY):
  Relevance:
```

**Additional Implementation Details**    In Section 3.2, we propose a technique for fitting the $\lambda$ parameter so as to minimize asymptotic variance. We remark that in our applications, we sometimes will clip $\lambda$ to enforce it within a range, for example, $[0, 1]$, to obtain more stable performance. Our theoretical results still hold since $\lambda$ will still converge to a constant.

## B. Additional Experiment Results

Figure 6 features coverage results for PovertyMap, GeneExpression, and OpinionQA. Given that we calculate 95% confidence intervals with QuEst in these experiments, we examine whether they contain the true quantity at least this often. Our empirical results empirically validate our confidence intervals.

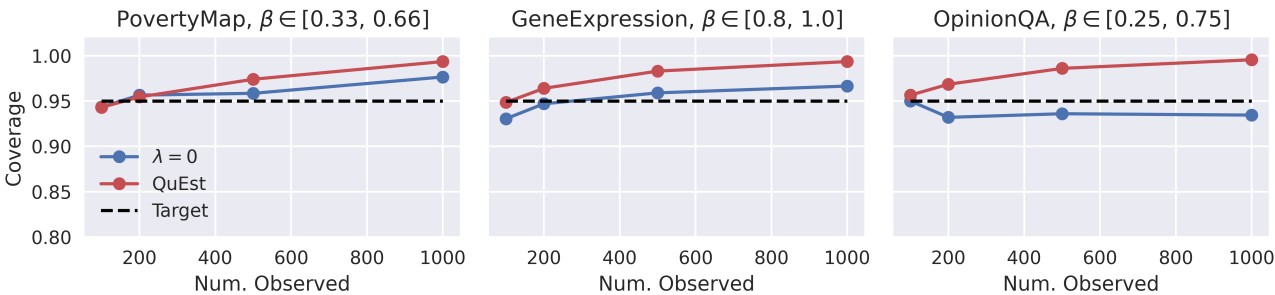

*Figure 6.* Coverage results across 3 datasets and using different numbers of observed data.

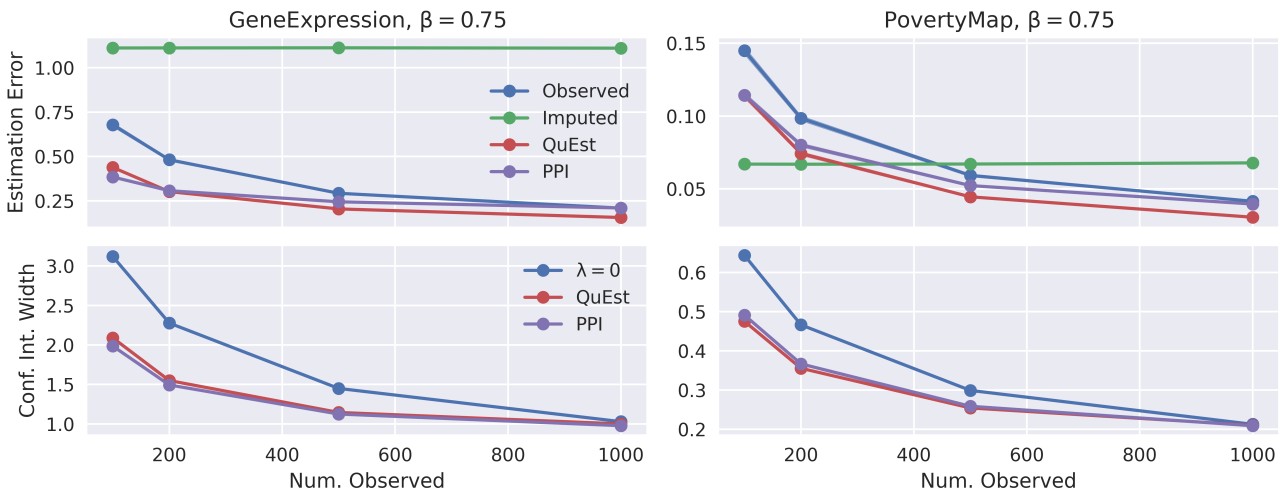

*Figure 7.* Experimental results for estimating VaR from three datasets (PovertyMap, GeneExpression, OpinionQA). **Top:** Estimation error vs. number of observed labels. **Bottom:** Confidence interval width vs. number of observed labels.

### B.1. QuEst Extension Experiments

To examine the potential benefits of directly optimizing a parameterized weighting function to estimate QBDMs, we construct a pathological case where a single correction for variance is unlikely to suffice: a dataset that exhibits *heteroskedasticity*, i.e. variance whose magnitude changes with data values. We sample 50000 linearly-spaced values from 1 to $v$, jitter these values by randomly adding or subtracting a number between 0 and $\delta$. We then add heteroskedasticity by first scaling data points proportionally to their absolute value, then normalizing all values to $[0, 1]$. We use a basis function that maps each point on the CDF to a combination of 30 sinusoids (similar to the positional embedding used in LLMs), and optimize a 30-dimensional vector to rectify the embedded CDF. Each experimental result is averaged over 100 trials with differing random seeds.

Figure 8 shows that QuEst-Opt successfully adapts to the heteroskedasticity when imputing the IQR (25th-75th percentile), outperforming QuEst across a range of observed dataset sizes. Notably, the improvement over QuEst is greatest in the ultra-low observed data regime. We observe a similar pattern in two real-world datasets (Figure 9). QuEst-Opt performs comparably to QuEst with a tuned $\lambda$ on Opinion QA, but performs better on the more complex Gene Expression dataset, particularly in the cases with very few observed samples.

## C. Additional Theory Results

In this section, we will demonstrate the omitted parts in our main context, including proofs and various of detailed formulas.

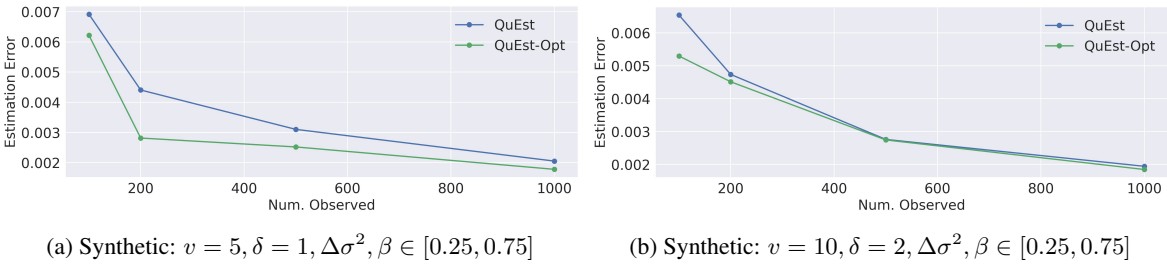

(a) Synthetic: $v = 5, \delta = 1, \Delta\sigma^2, \beta \in [0.25, 0.75]$      (b) Synthetic: $v = 10, \delta = 2, \Delta\sigma^2, \beta \in [0.25, 0.75]$

*Figure 8.* QuEst-Opt outperforms QuEst on synthetic datasets that exhibit heteroskedasticity (varying variance). Datasets constructed by sampling 50,000 values from 1 to $v$, jittering values randomly by at most $\delta$, and adding heteroskedasticity ($\Delta\sigma^2$).

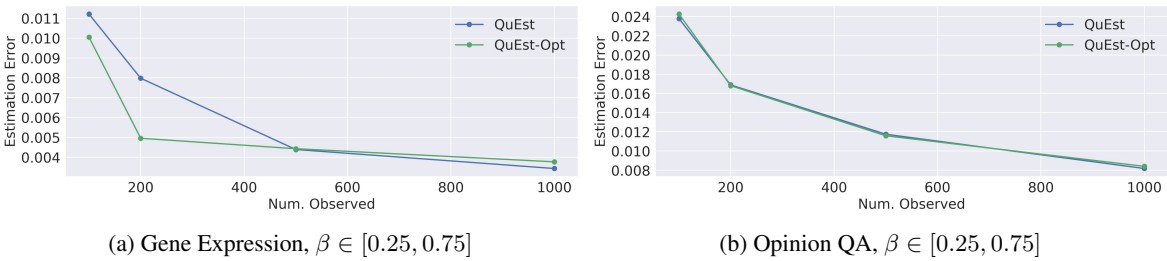

(a) Gene Expression, $\beta \in [0.25, 0.75]$      (b) Opinion QA, $\beta \in [0.25, 0.75]$

*Figure 9.* QuEst-Opt performs comparably to or better than QuEst on real-world datasets.

### C.1. Asymptotic Normality of $Q_\psi(F_n)$

In this subsection, we will establish central limit theorem (CLT) for $Q_\psi(F_n)$ for $\psi$ is a (1) bounded smooth function; (2) almost everywhere bounded and smooth function with finite discontinuous points; (3) Dirac delta function.

#### C.1.1. CLT FOR $Q_\psi(F_n)$ WITH BOUNDED SMOOTH $\psi$

First, by Theorem 22.3 in (Van der Vaart, 2000), we have that

$$\sqrt{n}\Big(Q_\psi(F_n) - \mathbb{E}Q_\psi(F_n)\Big) \to_D \mathcal{N}(0, \sigma_1^2(\psi, F)), \tag{3}$$

where

$$\sigma_1^2(\psi, F) = \int\int \big(F(u \wedge v) - F(u)F(v)\big)\psi(F(u))\psi(F(v))dudv.$$

Thus, if we can further prove that $\mathbb{E}Q_\psi(F_n) - Q_\psi(F) = o_p(1/\sqrt{n})$, then we can have conclusion that

$$\sqrt{n}\Big(Q_\psi(F_n) - Q_\psi(F)\Big) \to_D \mathcal{N}(0, \sigma_1^2(\psi, F)). \tag{4}$$

**Proof of $\mathbb{E}Q_\psi(F_n) - Q_\psi(F) = o_p(1/\sqrt{n})$.** In this part, we will prove that

$$\sqrt{n}(\mathbb{E}Q_\psi(F_n) - Q_\psi(F)) \to_n 0.$$

Notice that

$$Q_\psi(F) = \int_0^1 \psi(p)F^{-1}(p)dp = \int \psi(F(q))qdF(q).$$

As as result,

$$Q_\psi(F_n) = \int \psi(F_n(q))qdF_n(q) = \frac{1}{n}\sum_{i=1}^n \psi(F_n(M_i))M_i.$$

Furthermore, we have

$$
\begin{aligned}
\mathbb{E}Q_\psi(F_n) &= \mathbb{E}\Big[\psi(F_n(M_1))M_1\Big] \\
&= \mathbb{E}_{M_1}M_1\mathbb{E}_{\{M_p\}_{p\neq 1}}\Big[\psi(F_n(M_1))|M_1\Big] \\
&= \mathbb{E}_{M_1}M_1\mathbb{E}_{\{M_p\}_{p\neq 1}}\Big[\psi(\frac{1}{n}+\frac{\sum_{p>1}\mathbf{1}\{M_p\leq M_1\}}{n})|M_1\Big] \\
&= \int q\mathbb{E}_{\{M_p\}_{p\neq 1}}\Big[\psi(\frac{1}{n}+\frac{\sum_{p>1}\mathbf{1}\{M_p\leq q\}}{n})\Big]dF(q).
\end{aligned}
$$

Thus,

$$
\mathbb{E}Q_\psi(F_n)-Q_\psi(F)=\int q\mathbb{E}_{\{M_p\}_{p\neq 1}}\Big[\psi(\frac{1}{n}+\frac{\sum_{p>1}\mathbf{1}\{M_p\leq q\}}{n})\Big]dF(q)-\int q\psi(F(q))dF(q).
$$

Now the only thing we need to do is to study $\mathbb{E}_{\{M_p\}_{p\neq 1}}\Big[\psi(\frac{1}{n}+\frac{\sum_{p>1}\mathbf{1}\{M_p\leq q\}}{n})\Big]-\psi(F(q))$ and apply dominated convergence theorem.

$$
\begin{aligned}
\mathbb{E}_{\{M_p\}_{p\neq 1}}\Big[\psi(\frac{1}{n}+\frac{\sum_{p>1}\mathbf{1}\{M_p\leq q\}}{n})\Big]-\psi(F(q)) &= \psi'(F(q))\mathbb{E}_{\{M_p\}_{p\neq 1}}\Big(\frac{1}{n}+\frac{\sum_{p>1}\mathbf{1}\{M_p\leq q\}}{n}-F(q)\Big) \\
&+\frac{1}{2}\mathbb{E}_{\{M_p\}_{p\neq 1}}\psi''(s)\Big(\frac{1}{n}+\frac{\sum_{p>1}\mathbf{1}\{M_p\leq q\}}{n}-F(q)\Big)^2,
\end{aligned}
$$

where $s\in[0,1]$.

First,

$$
\mathbb{E}_{\{M_p\}_{p\neq 1}}\Big(\frac{1}{n}+\frac{\sum_{p>1}\mathbf{1}\{M_p\leq q\}}{n}-F(q)\Big)=\frac{1}{n}(1-F(q)).
$$

Second,

$$
\mathbb{E}_{\{M_p\}_{p\neq 1}}\psi''(s)\Big(\frac{1}{n}+\frac{\sum_{p>1}\mathbf{1}\{M_p\leq q\}}{n}-F(q)\Big)^2\leq C\mathbb{E}\Big(\frac{1}{n}+\frac{\sum_{p>1}\mathbf{1}\{M_p\leq q\}}{n}-F(q)\Big)^2=O\Big(\frac{1}{n}\Big)
$$

for $C=\sup_{s\in[0,1]}\psi(s)$.

To sum up, we prove that

$$
\mathbb{E}Q_\psi(F_n)-Q_\psi(F)=O\Big(\frac{1}{n}\Big).
$$

Thus, based on the previous results, we have the following theorem:

**Theorem C.1.** *Suppose that $EM^2<\infty$, $\psi$ is a bounded function that is twice continuously differentiable, then*

$$
\sqrt{n}\Big(Q_\psi(F_n)-Q_\psi(F)\Big)\to_D\mathcal{N}(0,\sigma_1^2(\psi,F)), \tag{5}
$$

*where*

$$
\sigma_1^2(\psi,F)=\int\int\big(F(u\wedge v)-F(u)F(v)\big)\psi(F(u))\psi(F(v))dudv.
$$

C.1.2. CLT FOR $Q_\psi(F_n)$ WITH ALMOST EVERYWHERE BOUNDED AND SMOOTH FUNCTION HAVING FINITE
  DISCONTINUOUS POINTS

The proof for this case will be similar as before. We specify a class of smooth functions $\{\psi_k\}_{i=1}^\infty$ such that

$$\lim_{k\to\infty} \psi_k \to \psi.$$

From our previous results, the CLT holds for each $\psi_k$. Then, we apply dominated convergence theorem and get our final result such that

$$\sqrt{n}\Big(Q_\psi(F_n) - Q_\psi(F)\Big) \to_D \mathcal{N}(0, \sigma_1^2(\psi, F)), \tag{6}$$

where

$$\sigma_1^2(\psi, F) = \int\int \big(F(u\wedge v) - F(u)F(v)\big)\psi(F(u))\psi(F(v))dudv.$$

C.1.3. CLT FOR $Q_\psi(F_n)$ WITH DIRAC DELTA FUNCTION

For the case that $\psi = \delta_\beta$, there have already been classic results built regarding the CLT of quantile functions.

**Theorem C.2.** *Suppose that $EM^2 < \infty$ and there exists density function $f$ for $M$ that is positive and smooth enough,*

$$\sqrt{n}\Big(F_n^{-1}(\beta) - F^{-1}(\beta)\Big) \to_D \mathcal{N}(0, \sigma_1^2(\delta_\beta, F)), \tag{7}$$

*where*

$$\sigma_1^2(\delta_\beta, F) = \frac{\beta(1-\beta)}{f^2(F^{-1}(\beta))}.$$

The only thing we need to verify is that $\sigma_1^2(\delta_\beta, F) = \frac{\beta(1-\beta)}{f^2(F^{-1}(\beta))}$ is also included in our general formula of variance, i.e.,

$$\int\int \big(F(u\wedge v) - F(u)F(v)\big)\delta_\beta(F(u))\delta_\beta(F(v))dudv = \frac{\beta(1-\beta)}{f^2(F^{-1}(\beta))}.$$

This indeed holds since:

$$
\begin{aligned}
\int\int \big(F(u\wedge v) - F(u)F(v)\big)\delta_\beta(F(u))\delta_\beta(F(v))dudv &= \int\int \beta(1-\beta)\delta_\beta(u)\delta_\beta(v)dF^{-1}(u)dF^{-1}(v) \\
&= \int\int \frac{\beta(1-\beta)\delta_\beta(u)\delta_\beta(v)}{f(F^{-1}(u))f(F^{-1}(v))}dxdy \\
&\left(\text{since } dF^{-1}(u) = \frac{1}{f(F^{-1}(u))}du\right) \\
&= \frac{\beta(1-\beta)}{f^2(F^{-1}(\beta))}.
\end{aligned}
$$

Combining all the above results, we know that the CLT holds for $\psi$ is a (1) bounded smooth function; (2) almost everywhere bounded and smooth function with finite discontinuous points; (3) Dirac delta function.

**C.2. Asymptotic Normality of $\hat{Q}_\psi(\lambda)$**

Notice that $\hat{Q}_\psi(\lambda)$ is a sum of three $L$-statistics, and two of them are dependent, namely, $Q_\psi(F_n)$ and $Q_\psi(\tilde{F}_n)$. We need to show that $Q_\psi(F_n) - \lambda Q_\psi(\tilde{F}_n)$ will converge to a normal distribution asymptotically. This will be straightforward following Section C.1 and the proof in (Van der Vaart, 2000). Specifically, following the proof in (Van der Vaart, 2000), we have that $Q_\psi(F_n) - \lambda Q_\psi(\tilde{F}_n)$ and $\int \sqrt{n}(F_n - F)(y)e_1(y)dy - \lambda\int \sqrt{n}(\tilde{F}_n - \tilde{F})(y)e_2(y)dy$ have the same asymptotically distribution for some deterministic functions $e_1$ and $e_2$. The latter one could be written into sum of i.i.d. random variables,

which will lead to CLT straightforwardly. Thus, we can obtain CLT for $\hat{Q}_\psi(\lambda)$ for any fixed $\lambda$. For any fixed $\lambda$, under the same conditions as in Section C.1, if $n/N \to r$ for some $r \geq 0$, we have

$$\sqrt{n}\left(\hat{Q}_\psi(\lambda) - Q_\psi(F)\right) \to_D \mathcal{N}(0, \rho_\psi^2(\lambda, F, \tilde{F})),$$

where

$$\rho_\psi^2(\lambda, F, \tilde{F}) = \lambda^2(1 + r)\sigma_\psi^2(\tilde{F}) + \sigma_\psi^2(F) - 2\lambda\eta_\psi(F, \tilde{F})$$

and $\eta_\psi(F, \tilde{F})$ is the covariance $\text{Cov}\left(Q_\psi(F), Q_\psi(\tilde{F})\right)$. Here,

$$\text{Cov}\left(Q_\psi(F), Q_\psi(\tilde{F})\right) = \int\int \psi(F(z))\psi(\tilde{F}(w))\left(\mathbb{E}\mathbf{1}\{Z \leq z, W \leq w\} - F(z)\tilde{F}(w)\right)dzdw$$

and $Z \sim F, W \sim \tilde{F}$. To sum up, we have the following theorem.

**Theorem C.3** (Restatement of Theorem 3.1). *For any fixed $\lambda$, under regularity conditions in Section C.1, if $n/N \to r$ for some $r \geq 0$, we have*

$$\sqrt{n}\left(\hat{Q}_\psi(\lambda) - Q_\psi(F)\right) \to_D \mathcal{N}(0, \rho_\psi^2(\lambda, F, \tilde{F})).$$

*where $\rho_\psi(\cdot, \cdot, \cdot)$ is*

$$\rho_\psi^2(\lambda, F, \tilde{F}) = \lambda^2(1 + r)\sigma_\psi^2(\tilde{F}) + \sigma_\psi^2(F) - 2\lambda\eta_\psi(F, \tilde{F})$$

*if $\psi$ is a (1) bounded smooth function; (2) almost everywhere bounded and smooth function with finite discontinuous points; (3) Dirac delta function.*

### C.3. Power-tuning with $\lambda$

The $\rho_\psi^2(\lambda, F, \tilde{F})$ can be further optimized by choosing $\lambda$ appropriately. Given that

$$\rho_\psi^2(\lambda, F, \tilde{F}) = \lambda^2(1 + r)\sigma_\psi^2(\tilde{F}) + \sigma_\psi^2(F) - 2\lambda\eta_\psi(F, \tilde{F})$$

is a quadratic function regarding $\lambda$, we can choose

$$\lambda^* = \frac{\eta_\psi(F, \tilde{F})}{(1 + r)\sigma_\psi^2(\tilde{F})}$$

to optimize the quadratic function.

However, when we construct confidence interval, we only have access to sample version of all types of quantities. So, we will choose $\lambda$ to optimize $\rho_\psi^2(\lambda, F_n, \tilde{F}_n, \tilde{F}_N^u)$, which gives us

$$\hat{\lambda} = \frac{\eta_\psi(F_n, \tilde{F}_n)}{(1 + n/N)\sigma_\psi^2(\tilde{F}_N^u)}.$$

Here,

$$\eta_\psi(F_n, \tilde{F}_n) = \int\int \psi(F_n(z))\psi(\tilde{F}_n(w))\left(\frac{1}{n}\sum_{i=1}^{n}\mathbf{1}\{Z_i \leq z, W_i \leq w\} - F_n(z)\tilde{F}_n(w)\right)dzdw$$

where $Z_i$'s are samples making up $F_n$ and $W_i$'s are samples making up $\tilde{F}_n$

Notice that $\hat{\lambda} \to_n \lambda^*$, so by Slustky's rule we have that

$$\rho_\psi^{-1}(\lambda, F_n, \tilde{F}_n, \tilde{F}_N^u)\sqrt{n}\left(\hat{Q}_\psi(\hat{\lambda}) - Q_\psi(F)\right) \to_D \mathcal{N}(0, 1).$$

This will give us Corollary 3.1.

### C.4. Multi-Dimensional QuEst

The multi-dimensional QuEst is straightforward to derive following the univariate case in Theorem 3.1 and Theorem 22.3 in (Van der Vaart, 2000).

We here just illustrate a bit more regarding the covariance terms in $\hat{V}$. Specifically,

$$\text{Cov}(Q_{\psi_i}(F_n), Q_{\psi_j}(\tilde{F}_n)) = \hat{\lambda}_i\hat{\lambda}_j\text{Cov}(Q_{\psi_i}(\tilde{F}_N^u), Q_{\psi_j}(\tilde{F}_N^u)) + \hat{\lambda}_i\text{Cov}(Q_{\psi_i}(\tilde{F}_N^u), Q_{\psi_j}(F_n)) - \hat{\lambda}_i\hat{\lambda}_j\text{Cov}(Q_{\psi_i}(\tilde{F}_N^u), Q_{\psi_j}(\tilde{F}_n))$$

$$+ \hat{\lambda}_j\text{Cov}(Q_{\psi_i}(F_n), Q_{\psi_j}(\tilde{F}_N^u)) + \text{Cov}(Q_{\psi_i}(F_n), Q_{\psi_j}(F_n)) - \hat{\lambda}_j\text{Cov}(Q_{\psi_i}(F_n), Q_{\psi_j}(\tilde{F}_n))$$

$$- \hat{\lambda}_i\hat{\lambda}_j\text{Cov}(Q_{\psi_i}(\tilde{F}_n), Q_{\psi_j}(\tilde{F}_N^u)) - \hat{\lambda}_i\text{Cov}(Q_{\psi_j}(\tilde{F}_n), Q_{\psi_j}(F_n)) + \hat{\lambda}_i\hat{\lambda}_j\text{Cov}(Q_{\psi_i}(\tilde{F}_n), Q_{\psi_j}(\tilde{F}_n)).$$

Notice that $\tilde{F}_N^u$ will be independent to $F_n$ and $\tilde{F}_n$, so we only need to calculate terms in the following forms.

$$\text{Cov}\big(Q_\psi(F_n), Q_{\tilde{\psi}}(\tilde{F}_n)\big) := \int\int \psi(F_n(z))\tilde{\psi}(\tilde{F}_n(w))\Big(\frac{1}{n}\sum_{i=1}^n \mathbf{1}\{Z_i \le z, W_i \le w\} - F_n(z)\tilde{F}_n(w)\Big)dzdw;$$

$$\text{Cov}\big(Q_\psi(F_n), Q_\psi(\tilde{F}_n)\big) := \int\int \psi(F_n(z))\psi(\tilde{F}_n(w))\Big(\frac{1}{n}\sum_{i=1}^n \mathbf{1}\{Z_i \le z, W_i \le w\} - F_n(z)\tilde{F}_n(w)\Big)dzdw;$$

and

$$\text{Cov}\big(Q_\psi(F_n), Q_{\tilde{\psi}}(F_n)\big) := \int\int \psi(F_n(z))\tilde{\psi}(F_n(w))\Big(F_n(z\wedge w) - F_n(z)F_n(w)\Big)dzdw.$$

### C.5. Omitted Details of Extension: A Better Rectified Estimator

Recall that we further modify our estimator $\hat{Q}_\psi(\lambda)$ to $\hat{Q}(\psi, \tilde{\psi})$, which is defined as

$$\hat{Q}(\psi, \tilde{\psi}) \triangleq Q_{\tilde{\psi}}(\tilde{F}_N^u) + \big(Q_\psi(F_n) - Q_{\tilde{\psi}}(\tilde{F}_n)\big). \tag{8}$$

In this paper, we choose $\tilde{\psi}(\cdot) = \psi_\xi(\cdot) \triangleq \xi^T\phi(\cdot)$, where $\phi(\cdot)$ is a multi-dimensional vector consists of basis functions and $\xi$ is a tuning parameter vector.

Following our previous theories, if $n/N \to r$ for some $r \ge 0$, then, for any fixed $\xi$, we have that

$$\sqrt{n}\big(\hat{Q}(\psi, \psi_\xi) - Q_\psi(F)\big) \to_D \mathcal{N}(0, \rho_\psi^2(\xi, F, \tilde{F}))$$

where

$$\rho_\psi^2(\xi, F, \tilde{F}) = (1+r)\int\int \big(\tilde{F}(u\wedge v) - \tilde{F}(u)\tilde{F}(v)\big)\psi_\xi(\tilde{F}(u))\psi_\xi(\tilde{F}(v))dxdy$$

$$+ \int\int \big(F(u\wedge v) - F(u)F(v)\big)\psi(F(u))\psi(F(v))dudv$$

$$- 2\int\int \psi(F(z))\psi_\xi(\tilde{F}(w))\Big(\mathbb{E}\mathbf{1}\{Z \le z, W \le w\} - F(z)\tilde{F}(w)\Big)dzdw$$

and $Z \sim F, W \sim \tilde{F}$.

Meanwhile, we define the consistent sample version of variance $\rho_\psi^2(\xi, F_n, \tilde{F}_n, \tilde{F}_N^u)$

$$\rho_\psi^2(\xi, F_n, \tilde{F}_n, \tilde{F}_N^u) = \Big(1 + \frac{n}{N}\Big)\int\int \big(\tilde{F}_N^u(x\wedge y) - \tilde{F}_N^u(x)\tilde{F}_N^u(y)\big)\psi_\xi(\tilde{F}_N(x))\psi_\xi(\tilde{F}_N(y))dxdy$$

$$+ \int\int \big(F_n(x\wedge y) - F_n(x)F_n(y)\big)\psi(F_n(x))\psi(F_n(y))dxdy$$

$$- 2\int\int \psi(F_n(z))\psi_\xi(\tilde{F}_n(w))\Big(\frac{1}{n}\sum_{j=1}^n \mathbf{1}\{Z_j \le z, W_j \le w\} - F_n(z)\tilde{F}_n(w)\Big)dzdw.$$

Here $Z_i$'s are samples making up $F_n$ and $W_i$'s are samples making up $\tilde{F}_n$. Then, we have

$$\nabla_\xi^2 \rho_\psi^2(\xi, F_n, \tilde{F}_n, \tilde{F}_N^u) = \left(1 + \frac{n}{N}\right) \int \int \left(\tilde{F}_N^u(x \wedge y) - \tilde{F}_N^u(x)\tilde{F}_N^u(y)\right)\phi(\tilde{F}_N^u(x))\phi^T(\tilde{F}_N^u(y))dxdy$$

It is not hard to show that $\int \int \left(\tilde{F}_N^u(x \wedge y) - \tilde{F}_N^u(x)\tilde{F}_N^u(y)\right)\phi(\tilde{F}_N(x))\phi^T(\tilde{F}_N(y))dxdy$ is the covariance matrix of the vector

$$\int \phi(p)(\tilde{F}_N^u)^{-1}(p)dp$$

by Theorem 22.3 in (Van der Vaart, 2000), which means $\rho_\psi^2(\xi, F_n, \tilde{F}_n, \tilde{F}_N^u)$ is semi-definite positive.

Thus, if we further add a penalty term and optimize

$$\rho_\psi^2(\xi, F_n, \tilde{F}_n, \tilde{F}_N^u) + \frac{\alpha}{2}\|\xi\|^2$$

with $\alpha > 0$, this optimization objective is strongly convex.

We denote

$$\hat{\xi} := \operatorname{argmin}_\xi \rho_\psi^2(\xi, F_n, \tilde{F}_n, \tilde{F}_N^u) + \frac{\alpha}{2}\|\xi\|^2.$$

We consider building CLT for $\hat{Q}(\psi, \psi_{\hat{\xi}})$. To simplify the notation, we denote

$$\hat{\theta}(\xi) = \hat{Q}(\psi, \psi_\xi)$$

for all $\xi$. We further denote

$$\xi^* := \operatorname{argmin}_\xi \rho_\psi^2(\xi, F, \tilde{F}) + \frac{\alpha}{2}\|\xi\|^2$$

Assume $\psi$ to be bounded and smooth enough almost everywhere (for example, twice continuously differentiable). By Talyor expansion, we have

$$\sqrt{n}\hat{\theta}(\hat{\xi}) = \sqrt{n}\hat{\theta}(\xi^*) + \sqrt{n}\nabla_\xi\hat{\theta}(\xi^*)^T(\hat{\xi} - \xi^*) + o_p(\frac{1}{\sqrt{n}}).$$

As long as we can show $\sqrt{n}\nabla_\xi\hat{\theta}(\xi^*)^T(\hat{\xi} - \xi^*) = o_p(1)$, we then can successfully build CLT because we will have

$$\sqrt{n}\hat{\theta}(\hat{\xi}) \sim_D \sqrt{n}\hat{\theta}(\xi^*),$$

where $\sim_D$ means same in distribution.

Notice that

$$\nabla_\xi\hat{\theta}(\xi^*) = \nabla_\xi Q_{\psi_{\xi^*}}(\tilde{F}_N^u) - \nabla_\xi Q_{\psi_{\xi^*}}(\tilde{F}_n) \to_p 0,$$

since $\tilde{F}_n$ and $\tilde{F}_N^u$ both converges to $\tilde{F}$. This means we only need to prove that

$$\sqrt{n}(\hat{\xi} - \xi^*) = O_p(1).$$

By the strong convexity of $\hat{\sigma}^2(\xi) := \rho_\psi^2(\xi, F_n, \tilde{F}_n, \tilde{F}_N^u) + \frac{\alpha}{2}\|\xi\|^2$, we have that

$$(\nabla\hat{\sigma}^2(\hat{\xi}) - \nabla\hat{\sigma}^2(\xi^*))^T(\hat{\xi} - \xi^*) \geq \alpha\|\hat{\xi} - \xi^*\|^2.$$

Thus, we can easily get

$$\|\nabla\hat{\sigma}^2(\xi^*) - \nabla\sigma^2(\xi^*)\| = \|\nabla\hat{\sigma}^2(\xi^*)\| \geq \alpha\|\hat{\xi} - \xi^*\|$$

since $\nabla\sigma^2(\xi^*) = 0$ and $\nabla\hat{\sigma}^2(\hat{\xi}) = 0$ by the first-order optimality condition.

Finally, it is straightforward to prove that

$$\nabla\hat{\sigma}^2(\xi^*) - \nabla\sigma^2(\xi^*) = O_p(1/\sqrt{n})$$

when we view the dimension of $\xi$ as a constant. Thus, we know that

$$\hat{\xi} - \xi^* = O_p(1/\sqrt{n}).$$

The proof is complete.

Thus, we have the folowing theorem.

**Theorem C.4** (Restatement of Theorem 5.1). *If $\psi$ is bounded and twice continuously differentiable almost everywhere, we have*

$$\rho_\psi^{-1}(\xi, F_n, \tilde{F}_n, \tilde{F}_N^u)\sqrt{n}\Big(\hat{Q}_\psi(\psi, \psi_{\hat{\xi}}) - Q_\psi(F)\Big) \to_D \mathcal{N}(0, 1).$$

*where $\hat{\xi} = \text{argmin}_\xi \, \rho_\psi^2(\xi, F_n, \tilde{F}_n, \tilde{F}_N^u) + \frac{\alpha}{2}\|\xi\|^2$ and $\alpha$ is any pre-specified positive constant.*

