# OpenReview forum: "QuEst: Enhancing Estimates of Quantile-Based Distributional Measures Using Model Predictions"
_ICML.cc/2025/Conference — ICML 2025 poster_

### Official Review · Reviewer_dnuP · 2025-02-25

**Overall Recommendation:** 3

**Summary:**

This paper introduces the QuEst method, which combines a few observed data with a large quantity of imputed data to derive enhanced estimates and reliable confidence intervals for quantile-based distribution metrics (QBDM). The method demonstrates its value in real-world applications, such as LLM Auto-Evaluation, with both theoretical and experimental evidence supporting its effectiveness.

## update after rebuttal
In the rebuttal, the authors clarified that their method achieves variance reduction without extra assumptions and degrades when imputation quality is low, addressing my main concern. Hence, I maintain a positive rating.

**Claims And Evidence:**

- The claim that “QuEst can provide more accurate quantile-based distribution metric (QBDM) estimates by integrating observed and model-predicted data” is substantiated by theoretical and experimental results.
- The theoretical and empirical findings support the claim that “QuEst performs well in multidimensional and complex scenarios.” However, additional experiments in high-dimensional settings would further validate the method's efficacy.

**Essential References Not Discussed:**

The paper thoroughly discusses previous studies. While no critical references appear missing, the possibility of omissions cannot be completely ruled out due to my limited familiarity with this area.

**Experimental Designs Or Analyses:**

The experimental design and analysis are sound and comprehensively validate QuEst’s performance. However, incorporating comparisons with more advanced methods, such as those discussed in Section 3.2, would enhance the paper's persuasiveness.

**Methods And Evaluation Criteria:**

The proposed method and evaluation criteria are reasonable and effectively address the research problem. QuEst combines observed and imputed data to provide more accurate QBDM estimates, and experimental results across various domains demonstrate its effectiveness.

**Other Comments Or Suggestions:**

There are some duplicated references that should be checked.

**Other Strengths And Weaknesses:**

**Strengths**:
- S1: The topic is valuable, especially the application to "LLM Auto-Evaluation."
- S2: The paper is well-written and easy to follow.
- S3: Experimental results support the proposed claims and demonstrate the method’s effectiveness.

**Weaknesses**:
- W1: More comparisons with advanced methods, like those in Section 3.2, would strengthen the paper.
- W2: The motivation and advantages of tuning a set of basis function coefficients rather than the original $\lambda$, as presented in Section 5 ("Extension"), need further discussion.

**Questions For Authors:**

**Questions**:
- Q1. Could the authors further discuss the requirements for imputed data to assist in estimating the target statistics? Intuitively, when imputed data significantly deviates from the true values, the assumptions needed for certain conclusions (e.g. variance reduction) may be violated, raising questions about the necessity of imputed data.
- Q2. I wonder whether the proposed estimator $\hat Q_{\psi}(\lambda)$ achieves asymptotic variance reduction while maintaining asymptotic unbiasedness without any additional assumption, compared to the previous estimator $Q_{\psi}(F_n)$. If additional assumptions are introduced, please discuss what they are and whether they hold.
- Q3. Could the authors briefly discuss the finite-sample properties of the proposed estimator?

**Relation To Broader Scientific Literature:**

The proposed QuEst is a rigorous extension of the Prediction-Powered Inference (PPI) framework. By introducing L-estimators, it improves the estimation of quantile-based metric, showcasing advantages across multiple application areas.

**Theoretical Claims:**

I have not verified the correctness of the theoretical claims in the paper.

---

> ### Author Rebuttal · Authors · 2025-03-31
>
> We thank the reviewer for the careful assessment of our work and the thoughtful suggestions.  Below, we respond to particular concerns.
>
>
> [Q1: Comparisons with more methods, such as those in Section 3.2]
>
> With respect to baselines, prior PPI variants are incompatible with the measures considered in our experiments; QuEst is the only existing specialized method for rigorously combining labeled and imputed data to estimate and provide CIs on these measures.
> We will clarify these choices in the final draft, specifying our exact baselines and emphasizing why specialized PPI methods do not directly apply to quantile-based distributional measures.
> This should ensure readers understand both the rationale behind our baseline selection and the distinct advantages offered by QuEst.
>
> Also, in order to further respond to these concerns, we have completed additional experiments on estimating the $\beta-VaR$ QBDM (Table 1).  This experiment does allow for a direct comparison to PPI, since it is estimating a single quantile.
> We find that QuEst produces higher quality estimates than our usual suite of observed and imputed baselines as well as the PPI method from Angelopoulos et al.  Representative figures will be included in the camera-ready version of the paper.
>
>  | Num. Data | Observed               | Imputed               | QuEst                 | PPI                   |
>  |-----------|------------------------|-----------------------|-----------------------|-----------------------|
>  | 250       | 9.2e-02 ± 6.7e-03      | **6.3e-02 ± 1.8e-03**     | 6.5e-02 ± 4.8e-03     | 6.7e-02 ± 4.9e-03     |
>  | 500       | 6.5e-02 ± 4.2e-03      | 6.6e-02 ± 1.9e-03     | **4.3e-02 ± 2.9e-03**     | 4.5e-02 ± 3.4e-03     |
>  | 1000      | 4.3e-02 ± 3.2e-03      | 6.6e-02 ± 1.7e-03     | **3.0e-02 ± 2.0e-03**     | 3.5e-02 ± 2.4e-03     |
>  | 1500      | 3.5e-02 ± 2.2e-03      | 6.6e-02 ± 2.0e-03     | **2.4e-02 ± 1.8e-03**     | 3.2e-02 ± 2.4e-03     |
>
>
> [Q2: The motivation and advantages of our extension in Section 5.]
>
> We here address the reviewer's question from two aspects:
>
> - Motivation: We leverage the fact that our estimator remains asymptotically unbiased regardless of which weighting function $\tilde{\psi}$ is chosen, because the expectations of its two terms cancel out. Thus, by letting $\tilde{\psi}$ be a flexible tuning function---rather than simply setting $\tilde{\psi} = \psi$---we can optimize for better debiasing and variance reduction.
>
> - Advantage: Instead of tuning a single scalar, we introduce a basis-function approach (e.g., polynomial or ``spline'' basis) and optimize the associated coefficient vector $\xi$. This advanced method, backed by new theorems and techniques, can yield more efficient estimators in both synthetic and real-world settings, as it provides added flexibility in controlling bias and variance.
>
>
> [Q3: Imputed data quality]
>
> With respect to imputation quality, we emphasize that QuEst’s adaptive nature ensures that it remains valid even if the imputed data deviates substantially from ground truth: if the imputed data are of high quality, then we choose a large $\lambda$ to weight more on the imputed data in the estimator. On the other hand, if the imputed data is of low quality, we will set smaller weight to imputed data (if $\lambda=0$, it just uses the gold-standard data alone). Since we are optimizing the tuning parameter $\lambda$, we would expect our estimator can always perform better than only using gold-standard label alone.
> Empirically (as in Fig. 4 with correlation $<0.1$), the estimator self-corrects by increasingly falling back on the smaller observed dataset if predictions are poor (with the $\alpha$ parameter).
>
>
> [Q4: Additional assumptions]
>
> As it pertains to assumptions, our estimator achieves asymptotic variance reduction relative to naive estimates without imposing additional assumptions beyond those enumerated in the paper. As mentioned in the previous question, our framework can automatically adapt to the quality of imputed data and  we would expect our estimator can always perform better than only using gold-standard label alone.
>
>
> [Q5: Finite-sample properties of the proposed estimator]
>
> There are two points we want to address. 1. Building central limit theorems actually only require a couple of hundreds gold-standard labeled data, so in practice, our framework works quite well with limited gold-standard labeled data. 2. If we falls into the regime of extreme small data, i.e., we only have less than 100, then the data uncertainty will be too huge and **no** statistically principled methods would work in the distribution-agnostic setting (this can be proved by using "no free-lunch theorem" in learning theory). We have to impose more assumptions to gain stronger results.

---

> > ### Comment · Reviewer_dnuP · 2025-04-04
> >
> > Thank you for your rebuttal, and I will maintain my positive score.

---

### Official Review · Reviewer_iSpq · 2025-03-08

**Overall Recommendation:** 3

**Summary:**

Authors propose a novel framework—QuEst—that enhances quantile based distributional measure estimation by combining scarce high quality observations with abundant model-predicted data. It produces an asymptotically unbiased estimator with reduced variance and can be further optimized using spline functions. Empirical results demonstrate that QuEst achieves accurate estimates and tighter confidence intervals  than previous methods in different domains.

## Update after rebuttal:
I thank the authors for their responses. My overall rating and assessment of the work remains positive, and so I maintain my score and recommend weak accept.

**Claims And Evidence:**

QuEst can achieve more accurate and reliable estimates for quantile￾based distributional measures than methods relying solely on either
observed or imputed data.

**Essential References Not Discussed:**

All good.

**Experimental Designs Or Analyses:**

- Overall, the experimental evaluation is thorough and well-designed, but there are a few areas worth noting: QuEst’s performance improves as the quality (i.e., the correlation between observed and imputed data) increases. However, the  experiments do not fully explore scenarios where imputed data is of lower quality or noisy.
- The method hinges on the proper tuning of the λ parameter (or its optimized variant in QuEst-Opt). While the paper provides closed-form expressions and some empirical evidence of optimal tuning, additional details on hyperparameter sensitivity and robustness across a wider
range of settings might further strengthen the evaluation

**Methods And Evaluation Criteria:**

The method and evaluation in the paper are well-motivated and comprehensive.

**Other Comments Or Suggestions:**

No

**Other Strengths And Weaknesses:**

- As the authors mentioned in the limitations section, a key weakness is the heavy reliance on high-quality imputed data, which may reduce robustness when predictions are noisy. However, this paper follows a similar style to PPI, which leverages pretrained model information and provides valid confidence intervals for estimating certain quantities. In this context, this approach is reasonable.
- In Figure 2, only the mean error is reported. Please include variance by running additional experiments.
- Does the prompt design impact the performance of the proposed method?
- Can more baseline methods be included? Currently, only two naive baselines are presented in the paper.

**Questions For Authors:**

No

**Relation To Broader Scientific Literature:**

The impact statement in this paper has well stated the borader impact of this paper.

**Theoretical Claims:**

The theoretical claims appear well-founded and are built upon established statistical principles.

---

> ### Author Rebuttal · Authors · 2025-03-31
>
> We thank the reviewer for the thoughtful remarks.  Please see below for responses to individual concerns.
>
> [Q1: Imputed data quality]
>
> First, we would like to clarify that our remark in the limitations section concerned the problem that **if we want to have significant gain in variance reduction**, we need **relatively** high-quality predictions. There are two points we want to address here regarding methodology:
>
> - We want to emphasize, **relatively** high-quality imputed data is enough to observe a significant gain in variance reduction. We don't need $100\%$ correct annotations for unlabeled data. Actually, in our experiments (Figure 4), we show that QuEst performs effectively with correlation of less than 0.1 (a very noisy regime) between the observed and imputed labels, and we still observe significant benefit by incorporating the imputed data in the estimator.
>
> - Another benefit of this framework is that our framework can automatically adapt to the quality of imputed data: if the imputed data are of high quality, then our algorithm will choose a large $\lambda$ to weight more on the imputed data in the estimator. On the other hand, if the imputed data is of low quality, our algorithm will set smaller weight to imputed data (if $\lambda=0$, it just uses the gold-standard data alone). Since our algorithm is optimizing the tuning parameter $\lambda$, we would expect our estimator can always perform better than only using gold-standard label alone.
>
>
>
>
> We will explicitly clarify this distinction in the final draft and emphasize that QuEst automatically debiases poor imputation quality and gracefully reverts to using the observed data as the primary signal.
>
>
> [Q2: Hyperparameters]
>
> Also, in response to the reviewer’s concern we will further clarify that our core estimator does not involve any hyperparameters in the closed-form tuning for $\lambda$, which we derived theoretically.  We believe that our empirical results across five distinct datasets and a wide range of labeled/unlabeled splits suggest this tuning is robust.
>
>
> [Q3: Variance in Figure 2]
>
> With respect to Figure 2, we ran 2000 trials, and the resulting variance in Figure 2 is extremely small, rendering the error bars nearly invisible for most of the plots.
>
>
> [Q4: Prompt design]
>
> Although we did not vary prompt design in our study, QuEst remains valid regardless of how the imputed data is generated—it will simply adapt to the correlations observed between imputed and true labels. Empirically, we show that even inaccurate predictions can be used effectively by QuEst: for example, accuracy on OpinionQA is roughly 50\%, but the QuEst estimate still consistently beats baselines.
>
>
> [Q5: Other baselines]
>
> With respect to baselines, prior PPI variants are incompatible with the measures considered in our experiments; QuEst is the only existing specialized method for rigorously combining labeled and imputed data to estimate and provide CIs on these measures.
> Thus we compared against the "classic" approach of using only the labeled data, as well as the use of imputed data only.
> For the confidence intervals, our "classic" baseline still requires our highly non-trivial CLT derivation (with $\lambda=0$).
> We will clarify these choices in the final draft, specifying our exact baselines and emphasizing why specialized PPI methods do not directly apply to quantile-based distributional measures. This should ensure readers understand both the rationale behind our baseline selection and the distinct advantages offered by QuEst.
>
> Also, in order to further respond to these concerns, we have completed additional experiments on estimating the $\beta-VaR$ QBDM (Table 1).  This experiment does allow for a direct comparison to PPI, since it is estimating a single quantile.
> We find that QuEst produces higher quality estimates than our usual suite of observed and imputed baselines as well as the PPI method from Angelopoulos et al.  Representative figures will be included in the camera-ready version of the paper.
>
>  | Num. Data | Observed               | Imputed               | QuEst                 | PPI                   |
>  |-----------|------------------------|-----------------------|-----------------------|-----------------------|
>  | 250       | 9.2e-02 ± 6.7e-03      | **6.3e-02 ± 1.8e-03**     | 6.5e-02 ± 4.8e-03     | 6.7e-02 ± 4.9e-03     |
>  | 500       | 6.5e-02 ± 4.2e-03      | 6.6e-02 ± 1.9e-03     | **4.3e-02 ± 2.9e-03**     | 4.5e-02 ± 3.4e-03     |
>  | 1000      | 4.3e-02 ± 3.2e-03      | 6.6e-02 ± 1.7e-03     | **3.0e-02 ± 2.0e-03**     | 3.5e-02 ± 2.4e-03     |
>  | 1500      | 3.5e-02 ± 2.2e-03      | 6.6e-02 ± 2.0e-03     | **2.4e-02 ± 1.8e-03**     | 3.2e-02 ± 2.4e-03     |

---

> > ### Comment · Reviewer_iSpq · 2025-04-01
> >
> > Thanks for your detailed response and I will keep my score.

---

### Official Review · Reviewer_bBBK · 2025-03-14

**Overall Recommendation:** 2

**Summary:**

The authors propose QuEst to estimate a quantile-based distributional measure (see Defintion 1) in a setting where there is a small set of high-quality data and a large set of low-quality data. For the high-quality data, it assumes that their low-quality estimates are also known and follow the same low-quality distribution that governers the low-quality data. In this setting, the shift between high- and low-quality distribution can be estimated using the small set of data, and then be used to correct the estimate computed using only the large set of low-quality data. This correction is the key step of their proposed QuEst, as described in Eq. (2). The hyperparameter $\lambda$ in Eq. (2) is optimized using Theorem 3.1, which essentially balances the contribution of the high- and low-quality data based on their proportions. By using the optimized $\lambda$ in Eq. (2), the authors arrive at their proposed QuEst.

## update after rebuttal
- Despite the heavy theoretical background, the high-level intuition seems simple to me. According to the assumptions made in Section 3.1, it is expected that $Q\_{\psi}(F_n) - \lambda Q\_{\psi}(\widetilde{F}_n) \approx Q\_{\psi}(F\_{N}\^{u}) - \lambda Q\_{\psi}(\widetilde{F}\_{N}^{u})$ because $\lim\_{n\to\infty}Q\_{\psi}(F_n) = \lim\_{N\to\infty}Q\_{\psi}(F\_{N}\^{u})$ and $\lim\_{n\to\infty}Q\_{\psi}(\widetilde{F}_n) = \lim\_{N\to\infty}Q\_{\psi}(\widetilde{F}\_{N}^{u})$, which is roughly an application of some variant of law of large numbers. Rearranging this approximate equality yields Eq. (2), which underpins the proposed method. After reading the first rebuttal, I realized that the primary difference lies in the specific variant of the law of large numbers employed. The authors distinguish the two variants using the concepts of M-estimators and L-estimators. However, I am uncertain about the novelty of this distinction in the present context.

- Regarding the spline-function-based method, I think the authors' initial rebuttal was weak, as the relevant content is entirely missing. After reading their second response (which contains information absent from the submission), I agree that using a function basis could potentially improve the quality of estimation. However, the submission lacks concrete theoretical support for evaluating its efficiency, and I do not find the experimental results in Table 3 sufficient to empirically validate the claimed superior efficiency. Moreover, how their spline-function-based method was implemented is also missing.

Overall, I will raise my score to 2. The main reason it is not higher is that the section on the spline-function-based method is clearly incomplete, considering that the authors highlight it as a key contribution in the submission.

**Claims And Evidence:**

One of the contributions they claim in the abstract is that "Further, we offer a novel spline function based method for optimizing our method." However, the word "spline" does not appear outside the abstract and the introduction section.

**Essential References Not Discussed:**

I am not familiar with the context.

**Experimental Designs Or Analyses:**

One issue is that they do not clearly define what the "classic method" is, which is used in the experiments comparing confidence intervals.
Another *possible* issue is the absence of other baselines in their experiments comparing the quality of estimates.

**Methods And Evaluation Criteria:**

The proposed QuEst makes sense in their specified setting.

**Other Comments Or Suggestions:**

NA

**Other Strengths And Weaknesses:**

The major concern I have is what is novel in this paper compared to the cited references. The key step of their proposed QuEst, as described in Eq. (2), already appears in (Angelopoulos et al. 2023, Section 2.1). Moreover, the idea of optimizing $\lambda$ in Eq. (2) can also be found in (Angelopoulos et al. 2023, Section 6). The only difference I noticed is the quantity to estimate.

**References**


Angelopoulos, A. N., Duchi, J. C., & Zrnic, T. Ppi++: Efficient prediction-powered inference, 2023. URL https://arxiv.org/abs/2311.01453.

**Questions For Authors:**

- For the proposed QuEst, what is novel compared to the theoretical framework established in (Angelopoulos et al. 2023)?

- It is possible that I missed it, but where is the proposed so-called spline function based method?

**Relation To Broader Scientific Literature:**

I am not familiar with the context.

**Theoretical Claims:**

I did not verify the proofs of their theoretical claims, as they fall outside my area of expertise.

---

> ### Author Rebuttal · Authors · 2025-03-31
>
> We thank the reviewer for their time and consideration in reviewing our work.  Here we respond to specific concerns raised in the review. We hope that our answers can resolve the misunderstanding and help to increase our score.
>
> [Q1: Novelty compared to existing work.]
>
> At the first glance, the surface form of Eq. (2) might resemble that of Angelopoulos et al. (2023). However, there is a significant difference between our frameworks.
>
> QuEst advances the prediction-powered inference paradigm beyond M-estimators (e.g., means, single quantiles) to fully encompass L-estimators, which can express diverse distributional measures like VaR, CVaR, and multi-quantile segments. This expansion requires new theoretical machinery, as our estimator cannot be straightforwardly written into sums of i.i.d. random variables. We rely on techniques from L-statistics, e.g., Hajek decomposition, and we also need to prove that $\mathbb{E}Q_\psi(F_n)-Q_\psi (F)=o_p(1/\sqrt{n})$ for certain $\psi$ functions, thereby establishing asymptotic normality under suitable conditions. Furthermore, QuEst integrates the concept of “power tuning” into this broader class of estimators and extends it with more sophisticated (spline-function-based) corrections in Sec.5, improving statistical efficiency in scenarios where a simple scalar multiplier would be insufficient. We also address multidimensional estimation, allowing simultaneous handling of multiple distributional measures (e.g., tail segments of multiple metrics) in a way that is not possible in prior PPI methods.
>
> From a practical standpoint, this generalization to L-estimators is essential for real-world tasks where the distributional properties themselves matter, such as identifying extreme behaviors or analyzing key population subgroups. QuEst provides asymptotically unbiased estimates and valid confidence intervals for quantities of substantial interest in domains like finance, public policy, and healthcare. Alongside the theoretical results, we present new variance formulas, covariance expressions, and performance bounds tailored to the L-estimator setting, offering a blueprint for rigorous inference on an array of tail or segment-based metrics. In sum, QuEst goes far beyond merely adapting an existing correction term; it is a comprehensive framework that systematically enables distributional inference where M-estimator-based prediction-powered approaches do not apply.
>
> [Q2: Use of splines in extension]
>
> Sorry for causing confusion. We are referring to the result in our “Extension” section. There, we introduce a more complicated version of power tuning instead of just tuning a multiplicative scalar lambda. Specifically, we consider tuning the coefficient vector for a given basis function in $\xi^T\phi(\cdot)$ (the tuning parameter is $\xi$ here). We specifically tried polynomial function basis, that is why we call this “spline function based”. Notice that it is a completely new and more advanced tuning method and we introduce new theorems and new techniques to get **more statistically efficient estimators**.
>
> [Q3: “Classic” baseline for confidence intervals]
>
> With respect to baselines, prior PPI variants are incompatible with the measures considered in our experiments; QuEst is the only existing specialized method for rigorously combining labeled and imputed data to estimate and provide CIs on these measures.
> Thus we compared against the "classic" approach of using only the labeled data, as well as the use of imputed data only.
> For the confidence intervals, our "classic" baseline still requires our highly non-trivial CLT derivation (with $\lambda=0$).
> We understand that this was not clear in the original submission, and would definitely clarify these details in our future revision.
>
> [Q4: Other baselines]
>
> In order to respond to these concerns, we have completed additional experiments on estimating the $\beta-VaR$ QBDM (Table 1).  This experiment does allow for a direct comparison to PPI, since it is estimating a single quantile.
> We find that QuEst produces higher quality estimates than our usual suite of observed and imputed baselines as well as the PPI method from Angelopoulos et al.  Representative figures will be included in the camera-ready version of the paper.
>
>  | Num. Data | Observed               | Imputed               | QuEst                 | PPI                   |
>  |-----------|------------------------|-----------------------|-----------------------|-----------------------|
>  | 250       | 9.2e-02 ± 6.7e-03      | **6.3e-02 ± 1.8e-03**     | 6.5e-02 ± 4.8e-03     | 6.7e-02 ± 4.9e-03     |
>  | 500       | 6.5e-02 ± 4.2e-03      | 6.6e-02 ± 1.9e-03     | **4.3e-02 ± 2.9e-03**     | 4.5e-02 ± 3.4e-03     |
>  | 1000      | 4.3e-02 ± 3.2e-03      | 6.6e-02 ± 1.7e-03     | **3.0e-02 ± 2.0e-03**     | 3.5e-02 ± 2.4e-03     |
>  | 1500      | 3.5e-02 ± 2.2e-03      | 6.6e-02 ± 2.0e-03     | **2.4e-02 ± 1.8e-03**     | 3.2e-02 ± 2.4e-03     |

---

> > ### Comment · Reviewer_bBBK · 2025-04-04
> >
> > Thanks for your detailed response, it addresses some of my concerns. Here are my additional concerns:
> >
> > - As suggested in the response, $\phi(\cdot)$ is supposed to be a polynomial function basis. I looked into Appendices C.4, C.5 and Section 5, I did not find a precise definition of $\phi(\cdot)$. In my view, it is still unclear what polynomial function basis is, how it contributes to the developed theory, and why it is supposed to be more efficient, as stated in lines 408-409 of the left column. Do the authors mean $\phi_{k}(p) = \psi(p)^{k}$ ?
> >
> > - Additionally, I noticed that the authors made an assumption in a proof; see line 1017.  Although Theorem 5.1 states, "Under certain regularity conditions," I think it would be clearer to explicitly state the assumption, especially since it is brief. Moreover, in Table 1, to my understanding, the second measure does not satisfy this assumption, which might be seen as a limitation of their developed theory.
> >
> > I am quite hesitant to raise my score because the part related to the so-called "spline function-based" approach seems incomplete. In particular, the authors claim this as one of their contributions in the abstract.

---

> > > ### Author Response · Authors · 2025-04-05
> > >
> > > We thank the reviewer for their continued engagement with our work.  We are glad that we were able to address your major concern with respect to the novelty of QuEst with respect to existing methods. The main significance of our paper is in generalizing PPI and existing techniques to L-estimators.  This innovation requires highly non-trivial derivations, and enables a wide range of important applications, including those highlighted in our comprehensive experiment section.  Given that the reviewer expresses no more concerns with respect to the main contribution of our idea, we respectfully and sincerely request that they reconsider their score of 1.
> > >
> > > We also thank the reviewer for their close reading of our extension section. While this is not part of our core QuEst method, we believe that this add-on also holds value in allowing for better estimates and tighter confidence intervals.  We agree that this section would benefit from enhanced clarity in a final draft, and plan to address this.  However, we do maintain that the algorithm as presented in this section is both complete and correct.  Please see below for responses to your particular concerns; hopefully these may also lead you to consider raising your score.
> > >
> > > **Q1**: Use of polynomial basis function.
> > >
> > > **A1**: Here we will clarify more about the polynomial function basis. We are sorry about the confusion caused, and will include further clarification in our revised version.
> > >
> > > 1. We would like to point out our extension in Section 5 is for **general** basis functions $\phi$ such that $\tilde{\psi}(\cdot)=\xi^T\phi(\cdot)$. Here, $\phi(\cdot)$ can be any basis function, like a Fourier basis or polynomial function basis. The proof of the theory \textbf{does not} dependent on the polynomial function basis, and it is just an instance of one choice of $\phi(\cdot)$.
> > >
> > > 2. Here, polynomial function basis is a commonly used terminology in mathematics, which means $\phi(x)=(1,x,x^2,\cdots,x^k)^T$, where $k$ is our choice. We specifically used that in our experiment in Appendix B, Figure 6.
> > >
> > > 3. Regarding efficiency, this is because we can specify $\phi(\cdot)$ to be complex enough so that optimizing $\tilde \psi$ can reach a lower variance. Notice as long as we choose $\phi$ to be a family of expressive enough basis functions, finetuning $\tilde \psi$ is more flexible than just finetuning a multiplicative parameter $\lambda$ as we mentioned in line 416. Note that by classic approximation theory, a polynomial function basis can approximate a wide class of functions very well if we choose $k$ large enough.
> > >
> > > **Q2**: Assumptions and application to measures in Table 1.
> > >
> > > **A2**: Thanks you for the suggestion regarding moving assumptions to the main paper. We have stated the clear conditions in the Appendix C.5; we will also include that in the main statement in the revised version. Previously, our goal given space constraints was to briefly summarize the argument in the main paper, and leave highly technical details for interested readers to the appendix. Based on the reviewer's concern, we can move some important details forward.
> > >
> > > Additionally, we note that we **did not** claim that our theory in section 5 can cover all the cases in Table 1. But our theory is general enough to cover CVaR and Interval VaR, which already give rise to a lot of interesting applications as highlighted throughout the paper. Most importantly, Section 5 is just an extension section, this is **not** our main contribution. Our main contribution is greatly generalizing PPI to L-estimators with new techniques and apply it to new interesting applications in genomics, social science, and LLM evaluation.
> > >
> > >
> > > Again, we thank the reviewer for the time and consideration in reviewing our work.  Once again, we are pleased that the reviewer seems satisfied with the novel contribution of our main QuEst method, and hope that we have been able to address your questions about the extension section, which we will thoroughly revise based on the reviewer's feedback.  We would greatly appreciate if the reviewer would reconsider their evaluation based on our discussion, and potentially also in light of the feedback of other reviewers.

---

### Official Review · Reviewer_ZoCm · 2025-03-19

**Overall Recommendation:** 3

**Summary:**

The authors introduce QuEst, a framework for estimating quantile-based distributional measures (e.g., VaR and CVaR in financial mathematics), which combines a smaller set of real data with a much larger set of output data from machine learning predictions, in a similar fashion as the prediction-powered inference framework introduced by Angelopoulos et al. (2023). The authors extend QuEst to be able to provide point estimates and prediction intervals in multivariate settings, further providing a more efficient implementation using a spline-function-based method. The authors showcase the efficacy of their approach in various real datasets, as well as in a red-teaming and news summarization experiments using various LLMs.

**Claims And Evidence:**

The claims made by the authors are supported by convincing evidence, as the experiments provide validation of their technique in various settings.

**Essential References Not Discussed:**

N/A

**Experimental Designs Or Analyses:**

I went through the experiments and their setup, which I think are solid and motivate the presented work well.

**Methods And Evaluation Criteria:**

I am not an expert in prediction-powered inference, but as far as my understanding goes the methods and evaluation criteria are sensible.

**Other Comments Or Suggestions:**

- Can the authors comment on whether it is fair to say that this work extends the work of PPI on M-estimators to L-estimators? (This seems to be implied by line 155, second column). Can it be generalized further than quantile-based distributional measures?
- The baselines in experiments are not clearly explained, which I appreciate is due to the lack of space. I suggest the authors to include a more detailed explanation in the final version of the paper.

Line 137. "M-estiamtors" -> "M-estimators"
Line 199, second column. "simultaneously" -> "simultaneously."

**Other Strengths And Weaknesses:**

Overall, I've found the paper to be well written and well structured, with the idea of QuEst providing an interesting approach of obtaining estimations of quantile-based estimation measures.
As I am not an expert in this literature, my main question is about the degree of innovation provided by the paper: after going through the manuscript, my understanding is that QuEst provides (a) a better alternative to point estimation to existing approaches of specific quantiles, (b) extends to measure beyond M-estimators and (c) further extends this setup to high-dimensional settings. If this is correct, how does the estimation of (a) compare with existing PPI approaches, like the ones mentioned in line 45 (second column)?

**Questions For Authors:**

Please see the questions above.

Overall I think this a good paper, with my overall recommendation being influenced by some of the points highlighted above and being outside of the prediction-powered inference literature.

(Finally, as a note outside this review, it is a shame that after using CVar as an example there was not a comparison of CVar from QuEst versus more traditional financial mathematics approaches, but I am sure someone might work on that once the paper is out.)

**Relation To Broader Scientific Literature:**

The authors do a good job citing relevant literature in Section 2.

**Theoretical Claims:**

I have not checked closely the proofs, just skimmed through it. The application of the limiting theorem from Van der Vaart (2000), which I am familiar with, seems sensible and I have not identified any glaring mistake in the proof structure and reasoning.

---

> ### Author Rebuttal · Authors · 2025-03-31
>
> We thank the reviewer for the thoughtful feedback and for recognizing the contribution of QuEst.  Below we address your concerns and questions.
>
> [Q1: Relation to existing approaches]
>
> This is correct. Existing PPI methods deal with estimators written in the form of sums of i.i.d. random variables. We strictly generalize it by moving beyond M-estimators to a new class of L-estimators (i.e., quantile-based distributional measures). Specifically, if we choose the weighting function as 1, it will become the mean estimator of a loss function, which is the objective studied in PPI. So the family of estimators we study  is a strict generalization of the previous result. The family of estimators we investigate are no longer straightforwardly in the form of sum of i.i.d. random variables as those studied in PPI, thus, it incurs new challenges and requires new techniques to build central limit theorems.
>
> For example, in Appendix C.1.1, in order to prove CLT, we need to prove  $\mathbb{E}Q_\psi(F_n)-Q_\psi (F)=o_p(1/\sqrt{n})$ for certain $\psi$ functions. Then, we need to use techniques in L-statistics such as Hajek decomposition to build the final CLT. In practice, this shift addresses important real-world needs, such as assessing tail behaviors, population subgroups, or other segments of interest that existing PPI methods do not handle. Our derivations yield asymptotically unbiased estimators and valid confidence intervals for these more complex target measures. Thus, QuEst closes a gap in the literature by providing a unified approach that not only captures means or single quantiles, but also richer distributional statistics of concern in diverse and important fields like biology, economics, and healthcare.
>
> [Q2: Handling L-Estimators]
>
> This is mostly correct but not precise. Because in line 155, what we mean is that if we choose the weighting function as a special one (constant function 1), it will become the mean estimator of a loss function, which is the objective studied in PPI. Instead of claiming we can handle all L-estimators, our claim is that the family of estimators we proposed for QBDMs falls into the category of L-estimators. As for further generalization over QBDM, we can indeed generalize a bit to handle a wider range of other L-estimators (more general weighting function, and special nonlinear functional of $Q$). But a main focus here is on quantile-based distributional measures, as they are especially relevant in high-stakes applications like tail risk analysis and policy decisions.
>
> [Q3: Clarifying baselines]
>
> For baselines, prior PPI variants are incompatible with the measures considered in our experiments; QuEst is the only existing specialized method for rigorously combining labeled and imputed data to estimate and provide CIs on these measures.
> Thus we compared against the "classic" approach of using only the labeled data, as well as the use of imputed data only.
> For the confidence intervals, our "classic" baseline still requires our highly non-trivial CLT derivation (with $\lambda=0$).
> We will clarify these choices in the final draft, specifying our exact baselines and emphasizing why specialized PPI methods do not directly apply to quantile-based distributional measures. This should ensure readers understand both the rationale behind our baseline selection and the distinct advantages offered by QuEst.
>
> To further address these concerns, we have completed additional experiments on estimating the $\beta-VaR$ QBDM (Table 1).  This experiment does allow for a direct comparison to PPI, since it is estimating a single quantile, and is thus the only QBDM computable with PPI.
> We find that QuEst produces higher quality estimates than our usual suite of observed and imputed baselines as well as the PPI method from Angelopoulos et al.  Representative figures will be included in the revised version of the paper.
>
>  | Num. Data | Observed               | Imputed               | QuEst                 | PPI                   |
>  |-----------|------------------------|-----------------------|-----------------------|-----------------------|
>  | 250       | 9.2e-02 ± 6.7e-03      | **6.3e-02 ± 1.8e-03**     | 6.5e-02 ± 4.8e-03     | 6.7e-02 ± 4.9e-03     |
>  | 500       | 6.5e-02 ± 4.2e-03      | 6.6e-02 ± 1.9e-03     | **4.3e-02 ± 2.9e-03**     | 4.5e-02 ± 3.4e-03     |
>  | 1000      | 4.3e-02 ± 3.2e-03      | 6.6e-02 ± 1.7e-03     | **3.0e-02 ± 2.0e-03**     | 3.5e-02 ± 2.4e-03     |
>  | 1500      | 3.5e-02 ± 2.2e-03      | 6.6e-02 ± 2.0e-03     | **2.4e-02 ± 1.8e-03**     | 3.2e-02 ± 2.4e-03     |
>
>
> [Q4: CVaR]
>
> We do not fully understand this point. Our focus on CVaR as one of our primary QBDMs of interest was inspired by its use in the financial mathematics literature. Our innovation applies to this measure; our "classic" baseline corresponds to its application based on that literature. To our knowledge, there are no other known techniques for combining gold-standard and imputed data to enhance CVaR estimates.

---

### Decision · Program_Chairs · 2025-05-01

**Decision:**

Accept (poster)

**Comment:**

This paper introduces QuEst, a framework that combines scarce observed data with abundant imputed data for the estimation of quantile-based distributional measures (QBDMs). The reviewers generally found the topic valuable and the core idea of QuEst interesting, particularly for applications in LLM auto-evaluation and other domains where understanding distributional properties is crucial. The proposed method extends and generalizes the Prediction-Powered Inference (PPI) framework, expanding its applicability from M-estimators to a particular class of L-estimators. I recommend acceptance of this paper.

To further strengthen the paper, the authors should adequately address the feedback provided by the reviewers in their camera-ready version, including novelty clarification, experimental baselines, spline-based optimization method, imputed data quality, etc.